# Accelerating Block Coordinate Descent for LLM Finetuning via Landscape Expansion

**Qijun Luo**[1†]   **Yifei Shen**[2*]   **Liangzu Peng**[3]   **Dongsheng Li**[2]   **Xiao Li**[1*]

[1]The Chinese University of Hong Kong, Shenzhen

[2]Microsoft Research Asia

[3]University of Pennsylvania

## Abstract

Finetuning large language models (LLMs) is a resource-intensive task for researchers in academia, with memory constraints posing a key bottleneck. A classic optimization method, block coordinate descent (BCD), significantly reduces memory cost by segmenting the trainable parameters into multiple blocks and optimizing one active block at a time while freezing the others. However, we identify that blindly applying BCD to train LLMs can be inefficient for two reasons. First, optimizing only the active block requires backpropagating through multiple deeper yet inactive blocks, resulting in wasteful computations. Second, the frozen blocks, when they are not quite close to optimality, can narrow the optimization landscape, potentially misguiding the training of the active block. To address these issues simultaneously, we propose integrating BCD with *landscape expansion*, which unfreezes the inactive blocks and updates them in a cost-efficient manner during the same backpropagation as the update to the active block. Experiments on 8B and 70B models demonstrate that our proposed method surpasses memory-efficient baselines and matches Adam's downstream performance while requiring only 24 GB of memory for the 8B model and 300 GB for the 70B model.

## 1   Introduction

Large language models (LLMs) have gained significant popularity within the research community and industry. To drive progress in this field and satisfy a wide range of application needs, researchers most commonly finetune LLMs on diverse datasets tailored to various tasks and objectives. However, finetuning LLMs demands extensive computational resources, with memory being a primary constraint. For instance, optimizing a model with $N$ billion parameters using Adam requires at least $18N$ gigabytes of GPU memory (see Section 2). This memory limitation often prevents researchers from experimenting with larger models.

To address this practical challenge, researchers have developed memory efficient algorithms for LLM finetuning such as parameter efficient finetuning (PEFT), including Adapter [11], LoRA [13], prompt tuning [16], prefix tuning [17], etc. These techniques focus on training a small set of additional parameters while maintaining the original pretrained model unchanged. Other memory efficient methods for full parameter training have also been investigated. For example, Galore [43] applies a low-rank space projection to both the gradient and the optimizer's states to reduce memory consumption. For more related works, please refer to Section A.

---

[†]Work done during the internship at Microsoft Research Asia.

[*]Correspondence to Yifei Shen `<yifeishen@microsoft.com>`, Xiao Li `<lixiao@cuhk.edu.cn>`.

In addition to the existing approaches, a classic optimization paradigm, known as *block coordinate descent* (BCD), holds a strong potential for memory efficient LLM finetuning. When optimizing a model with $N$ billion parameters, BCD partitions the model parameters into $D$ blocks and optimizes over only one active block at a time. Consequently, the memory is decreased from $18 N$ to $2N + \frac{16N}{D}$ GB, as only the gradient and optimizer states of the active block need to be stored. In practice, BCD has been the method of choice in many data science problems in the last decade, with a wide array of variants developed for improving memory, performance, convergence, and efficiency; see, e.g., [12, 2, 41, 36, 33].

In stark contrast to the previous memory efficient methods, and despite its intuitive memory benefits, BCD has been overlooked and rarely explored in the context of deep neural networks (hence LLMs). It was not until very recently that [21] proposed BAdam, which integrates BCD into an LLM finetuning framework by training each active block with several Adam steps. Even though BAdam has shown preliminary success in reducing memory cost during training and improving performance at test time, its direct use of vanilla BCD leaves at least two fundamental aspects to be questioned:

- Computing the (stochastic) gradient for a *single* active block via backpropagation necessitates calculating the partial derivatives of the activations of *multiple* deeper yet inactive layers. This is wasteful of computation, since these partial derivatives are not used to update their corresponding weights.

- Given that the training objective is highly nonconvex, and since all blocks are frozen except for the active one, BCD tends to be misled by *its local view of the optimization landscape*, which potentially slows down its convergence speed.

Building on the foundation established by BAdam, we propose a simple remedy to the above two issues. Our new algorithmic framework, termed BREAD, is a blend of two components: (1) Similarly to BAdam, we update the active block using several Adam steps (the BCD component); (2) differently, we unfreeze the inactive blocks and update them using lightweight memory efficient optimization techniques (the *landscape expansion* component). Since landscape expansionutilizes the gradients of the activations that are already calculated, it adds minimal additional computation and in fact addresses the wasteful computation issue. Furthermore, the landscape expansioncomponent provides BCD a better view of the optimization landscape for better updating the current active block, thereby addressing the second point of concern. In combination, BREAD maintains the memory efficient feature of BCD with improved learning capability and faster convergence. Our main contributions are outlined as follows:

- **Limitations of Standard BCD in LLMs:** Our research identifies two fundamental limits of vanilla BCD when applied to LLMs: The wasted gradient computation during backpropagation and the suboptimal landscape caused by freezing inactive blocks. These limitations partly explain why the application of BCD in neural networks is uncommon.

- **Blending BCD with Landscape Expansion:** We propose a new algorithmic framework termed BREAD, which combines BCD with a landscape expansiontechnique to address these two limitations simultaneously. It unfreezes some of (or all) the inactive blocks and updates them using memory efficient optimization techniques. BREAD maintains the memory efficiency of BCD with improved optimization ability.

- **Excellent Performance:** Our experiments on instruction tuning and preference optimization with the Llama 3.1-8B and Llama 3.1-70B models demonstrate that BREAD clearly outperforms state-of-the-art memory efficient training methods and achieves comparable downstream performance to that of Adam on five math benchmarks and MT-bench scores.

## 2 Preliminaries on Block Coordinate Descent for LLM Training

Our main focus lies in improving the efficiency of BCD for finetuning LLMs. Therefore, we first review some preliminary concepts of LLM and BCD in this section.

**Objective of training LLMs.** Consider minimizing a general objective function $\min_{\boldsymbol{W}} H(\boldsymbol{W}) = \frac{1}{n} \sum_{j=1}^{n} h_j(\boldsymbol{W})$, where $\boldsymbol{W} \in \mathbb{R}^d$, $n$ is the number of data samples. In the context of training/finetuning LLMs, $h_j(\boldsymbol{W})$ represents the negative log-likelihood of the autoregressive probability $\mathbb{P}_{\boldsymbol{W}}[\boldsymbol{y}_j | \boldsymbol{x}_j] = \prod_{s=1}^{m} \mathbb{P}_{\boldsymbol{W}}[\boldsymbol{y}_{j,s} | \boldsymbol{y}_{j,1:s-1}, \boldsymbol{x}_j]$ for the $j$-th prompt $\boldsymbol{x}_j$ and its corresponding $j$-th output

$\boldsymbol{y}_j$. In most LLM models, this autoregressive probability is modeled by a transformer architecture [37], and thus $\boldsymbol{W} \in \mathbb{R}^d$ encompasses all trainable parameters of the transformer, including the query, key, value, and output attention matrices, as well as the gate, up, and down projection matrices of each transformer layer.

**BCD for LLM training.** BCD first splits the model parameters into $D$ blocks, i.e., $\boldsymbol{W} = \{\boldsymbol{W}_1, \cdots, \boldsymbol{W}_\ell, \cdots, \boldsymbol{W}_D\}$, where $\boldsymbol{W}_\ell \in \mathbb{R}^{d_\ell}$ and $\sum_{\ell=1}^{D} d_\ell = d$. The block partition in such a splitting can be very flexible. For instance, the block variable $\boldsymbol{W}_\ell$ can be either a single matrix or all the trainable matrices of a transformer layer. Then, at each block iteration, BCD updates only one active block while fixing the others at their most up-to-date values. This makes each sub-problem of BCD a $D\times$ smaller problem compared to the original one if the $D$ blocks are partitioned evenly. Suppose at the $(t+1)$-th block iteration the active block is $\boldsymbol{W}_\ell$, BCD optimizes the following problem:

$$\min_{\boldsymbol{W}_\ell \in \mathbb{R}^{d_\ell}} h(\boldsymbol{W}_1^{t+1}, \cdots, \boldsymbol{W}_{\ell-1}^{t+1}, \boldsymbol{W}_\ell, \boldsymbol{W}_{\ell+1}^t, \cdots, \boldsymbol{W}_D^t),$$

where $\mathcal{N} \subseteq \{1, \cdots, n\}$ is a batch of the training dataset. Updating from block $\ell = 1$ to block $\ell = D$ is counted as one block-epoch. Since it is intractable to solve (1) exactly, one can instead approximate the solution by implementing $K$ Adam steps, as utilized in BAdam [21].

**Memory efficiency of BCD.** We analyze the memory consumption induced during training under the mixed precision training setting [25]. The memory cost is attributed to the storage of the model parameters, gradients, and optimizer states. We consider an LLM with $N$ billion parameters and express GPU memory consumption in gigabytes (GB). Initially, one must store the FP16 model parameters for the backpropagation (BP) process, requiring $2N$ memory. Additionally, the optimizer maintains a copy of the model in FP32 precision, consuming another $4N$ memory. The gradients, momentum, and second moment vectors are all stored in FP32 precision with each requiring $4N$ memory. Consequently, the total memory required is at least $18N$. For example, in order to train a Llama 3-8B or a Llama 3-70B model, Adam requires at least $144$ GB or $1260$ GB of GPU RAM, respectively, which can be prohibitive in limited memory scenarios.

In sharp contrast to Adam, BCD only requires storing the FP32 model parameters, gradients, and optimizer states for the *active block* $\boldsymbol{W}_\ell$, which is only $1/D$ of the memory consumption needed for all the parameters. Thus, in addition to maintaining an FP16 model that requires $2N$ memory, BCD needs a total of only $2N + \frac{16N}{D}$ memory. Therefore, for training a Llama 3-8B or a Llama 3-70B model and when $D = 32$ or $D = 80$ (partition each transformer layer as a block), BCD only needs roughly $20$ GB or $154$ GB of GPU RAM, respectively, which is significantly cheaper compared to the costs of Adam. We refer to [21] for a more detailed analysis on memory cost.

## 3 Limitations of BCD for Neural Networks

In this section, we show that while BCD is memory-efficient for training LLMs, there are two major limitations when applying BCD for neural network training. To ease our analysis, let us consider a $L$-layer feedforward neural network model:

$$\boldsymbol{z}_{\ell+1} = f_{\boldsymbol{W}_\ell}^\ell(\boldsymbol{z}_\ell), \ \forall 1 \leq \ell \leq L, \quad \text{with} \quad \boldsymbol{z}_1 = \boldsymbol{x}, \tag{1}$$

where $L$ is the total number of layers, $\boldsymbol{x}$ is the input, $f_{\boldsymbol{W}_\ell}^\ell$ is the $\ell$-th layer's transformation.

**Limitation I: Ineffective utilization of intermediate derivatives during backpropagation.** Due to the compositional structure of deep neural networks, taking the stochastic gradient of the $\ell$-th layer's parameters $\boldsymbol{W}_\ell$ requires computing the partial derivatives with respect to all the activation values of deeper layers, as shown in the following equation:

$$\frac{\partial H}{\partial \boldsymbol{W}_\ell} = \underbrace{\frac{\partial H}{\partial \boldsymbol{z}_{L+1}} \frac{\partial \boldsymbol{z}_{L+1}}{\partial \boldsymbol{z}_L} \cdots \frac{\partial \boldsymbol{z}_{\ell+2}}{\partial \boldsymbol{z}_{\ell+1}}}_{I_{\ell+1}} \frac{\partial \boldsymbol{z}_{\ell+1}}{\partial \boldsymbol{W}_\ell}, \tag{2}$$

where $H$ is the objective function of the neural network training problem. During the backpropagation process, optimization methods such as Adam utilizes all the intermediate partial derivatives $I_{\ell+1}$ of the activations in (2) for computing the gradients of the $L, L-1, \cdots, \ell+1$-th layers' weight parameters. However, since BCD only updates the active block $\boldsymbol{W}_\ell$, the term $I_{\ell+1}$ is merely used for

calculating the gradient of $\boldsymbol{W}_\ell$, resulting in ineffective utilization of the computed partial derivatives of the activations during backpropagation.

**Limitation II: Convergence slowdown of BCD's sub-problem.** To tackle a training problem, optimization methods such as Adam optimize over all the trainable parameters $\boldsymbol{W}$, while the BCD optimization scheme (1) minimizes the objective over only the current active block, keeping the others fixed. Intuitively, BCD appears to narrow the optimization landscape of the training problem by freezing most of the parameters in each of its sub-problems, potentially eliminating better search directions that can lead to rapid decrease of the objective function.

To establish such an intuition formally, let us consider the following optimization problem:

$$\min_{\boldsymbol{W}_1,\cdots,\boldsymbol{W}_L,\boldsymbol{W}_{\text{out}}} H(\boldsymbol{W}) := F\left(f_{\boldsymbol{W}_{\text{out}}} \circ f^L_{\boldsymbol{W}_L} \circ f^{L-1}_{\boldsymbol{W}_L-1} \circ \cdots f^1_{\boldsymbol{W}_1}(x), y\right), \qquad (3)$$

where $x$ and $y$ are the input and the label, respectively. $f^\ell_{\boldsymbol{W}_\ell}$ is the transformation of $\ell$'s layer, which can be either a transformer layer or a feedforward layer. $f_{\boldsymbol{W}_{\text{out}}}$ is the transformation of output layer, e.g. the language modeling head of a transformer model. The following proposition establishes the condition where BCD's sub-problem is *strictly* suboptimal to the original problem.

**Proposition 3.1.** *(BCD's sub-problem may not include optima) Consider the objective (3) with $F(\cdot)$ being cross-entropy loss employed in the training of transformer model. When $\boldsymbol{W}_{out}$ is not of full rank, there exists a pair of $(x, y)$ such that BCD's sub-problem is strictly suboptimal to the original problem:*

$$\min_{\mathcal{S} \subseteq \{\boldsymbol{W}_1,\cdots,\boldsymbol{W}_L\}} H(\boldsymbol{W}_1,\cdots,W_L,W_{out}) > \min_{\boldsymbol{W}_1,\cdots,\boldsymbol{W}_L,\boldsymbol{W}_{out}} H(\boldsymbol{W}_1,\cdots,W_L,W_{out}). \qquad (4)$$

We remark that the above inequality is different from a direct conclusion of expressivity power: $\min_{\boldsymbol{W}_1} H(\boldsymbol{W}) \geq \min_{\boldsymbol{W}_1,\boldsymbol{W}_2} H(\boldsymbol{W})$, which does not induce the *strict* suboptimality. In Section D.3, we show that the strict suboptimality issue also exists for regression problem with $\ell_2$ loss under relatively mild assumptions.

The inequality (4) characterizes the suboptimality induced by freezing the output layer. We provide further analysis and empirical evidence on the effect of freezing intermediate layers in Section D.3.

**Possible slowdown in convergence.** Theorem 3.1 is only for a sub-problem of BCD, which does not imply that BCD will finally fail to converge. However, the *strict* inequality proven in Theorem 3.1 implies that BCD's sub-problem may not have the full optimization landscape for some concrete transformer problems, as the optimal solutions are not covered. This narrow landscape issue will potentially slow down the convergence of BCD's sub-problem to the optima, as the search direction towards the optimal solutions is excluded.

To address above limitations, one immediate approach is to apply Adam to inactive blocks as well. However, this essentially reverts to using Adam and undermines the memory efficient property of the BCD optimization scheme. In the next section, we will present several memory-efficient approaches to address them.

# 4   BREAD Framework and Its Analysis

In this section, we present our framework **B**lock coo**R**dinate d**E**scent via l**AnD**scape expansion (BREAD), which solve the two limitations revealed in Section 3 simultaneously, with almost negligible additional memory cost.

## 4.1   Motivations: Low-rank Expansion Addresses Limitations

The limitations in Section 3 attributes to BCD's design of updating only one block parameters at a time, primarily for the concern of memory-efficiency. In this section, we demonstrate that both limitations can be effectively addressed by applying low-cost updates on inactive blocks, with only negligible additional memory cost incurred. Formally, our approach is motivated by the following proposition.

**Proposition 4.1.** *(Rank-1 Expansion) Under the same assumption as Theorem 3.1, the suboptimality issue can be resolved by introducing rank-1 update on $W_{out}$:*

$$\min_{\mathcal{S} \cup \boldsymbol{C}} H(\boldsymbol{W}_1,\cdots,W_L,W_{out} + \boldsymbol{C}) = \min_{\mathcal{S} \cup \boldsymbol{W}_{out}} H(\boldsymbol{W}_1,\cdots,W_L,W_{out}),$$

*where $\mathcal{S} \subseteq \{W_1, \cdots, W_L\}$ is the active block parameter, and $C$ is a rank-1 matrix.*

Motivated by the above proposition, we develop the BREAD framework for accelerating BCD by performing low-cost update on inactive blocks with almost negligible additional memory cost.

## 4.2 The BREAD Framework

---

**Algorithm 1** BREAD-LoRA

---

1: **Input:** Model parameters $\{W_\ell^0\}_{\ell=1}^L$, number of blocks $D$, iterations per block $K$, training dataset $\mathcal{D} = \{(x_j, y_j)\}_{j=1}^n$, batch size $B$.
2: **Initialization:** Block-epoch index $t \leftarrow 0$, inactive block LoRA matrices $U_j^0, V_j^0$ and optimizer states $\tilde{s}_j^0 \leftarrow \mathbf{0}, \forall j \in [P]$.
3: **while** stopping criterion not met **do**
4:     generate a block partition $\pi = \{\pi_1, \ldots, \pi_D\}$;
5:     **for** one *block-epoch* $i = 1$ to $D$ **do**
6:         Determine partial updates or full updates $J \subset [P]$ as in (6);
7:         $s_{\pi_i}^{t,0} \leftarrow \mathbf{0}$;       // Re-initialize optimizer states for the active block
8:         $W_{\pi_i}^{t,0} \leftarrow W_{\pi_i}^t; \tilde{s}_J^{t,0} \leftarrow \tilde{s}_J^t$;
9:         **for** *landscape expansion block updates* $k = 1$ to $K$ **do**
10:             sample a data batch in random-reshuffled order $\mathcal{D}_B = \{(x_j, y_j)\}_{j=1}^B \sim \mathcal{D}$;
11:             **within** one backward pass on the data batch $\mathcal{D}_B$ **do**
12:                 compute the active block's grad. $g_i^{t,k}$ and correction matrices' grad. $\tilde{g}_J^{t,k}$;
13:             **end within**
14:             // Update active block and correction matrices
15:             $W_{\pi_i}^{t,k}, s_{\pi_i}^{t,k} \leftarrow \mathsf{AdamStep}(W_{\pi_i}^{t,k-1}, g_{\pi_i}^{t,k}, s_{\pi_i}^{t,k-1})$;
16:             $U_J^{t,k}, V_J^{t,k}, \tilde{s}_J^{t,k} \leftarrow \mathsf{AdamStep}(\{U_J^{t,k-1}, V_J^{t,k-1}\}, \tilde{g}_J^{t,k}, \tilde{s}_J^{t,k-1})$;
17:         **end for**
18:         $W_{\pi_i}^{t+1} \leftarrow W_{\pi_i}^{t,K}; U_J^{t+1} \leftarrow U_J^{t,K}; V_J^{t+1} \leftarrow V_J^{t,K}; \tilde{s}_J^{t+1} \leftarrow \tilde{s}_J^{t,K}; s_{\pi_i}^{t,K} \leftarrow \mathbf{None}$;
19:     **end for**
20:     $t \leftarrow t + 1$
21: **end while**
22: **return** parameters $\{W_\ell^t\}_{\ell=1}^L$ and correction matrices $\{C_j^t\}_{j=1}^P$

---

Similar to BCD framework, BREAD splits the model into $D$ blocks, which can be partitioned either in a layer-wise or matrix-wise manner. Then, each block sub-problem is approximately solved using $K$ steps. Importantly, BREAD not only optimizes the active block as in BCD, but also the weights of inactive blocks for better optimization landscape in a memory-efficient manner. We introduce two concrete algorithmic instances for updating inactive blocks with **almost negligible additional memory cost**.

**Algorithm I: Low-rank based expansion.** Motivated by Theorem 4.1, we propose to introduce additional *trainable low-rank expansion matrices* to inactive blocks $\{W_{\ell'}\}_{\ell' \neq \ell}$, where $W_\ell$ is the current active block. The simplest implementation of this idea is to add LoRA with extremely low rank (e.g., rank-4) to inactive blocks.

$$W_{\ell'} + U_{\ell'} V_{\ell'}, \quad U_{\ell'} \in \mathbb{R}^{m \times r}, V_{\ell'} \in \mathbb{R}^{r \times n}, \quad \forall \ell' \neq \ell. \tag{5}$$

We present the detailed procedure in Algorithm 1. For each landscape corrected update (Algorithm 1 line 9–16), we sample a batch of data in a random reshuffled manner, and calculate the gradient of both active block and the expansion matrices within one backward pass. Then, we update the active block and expansion matrices with a single Adam step. The optimizer states of the expansion matrices are accumulated throughout the entire algorithm execution, as they occupy only negligible memory space.

In the subsequent sections, we refer the Algorithm 1 as **BREAD-LoRA**. Motivated by the [7], we also propose a variant that uses higher learning rate for $U$, i.e. $\frac{\alpha_U}{\alpha_V} \gg 1$, which we refer as **BREAD-LoRA+**. We remark that Algorithm 1 naturally allows any low-rank based memory efficient methods for making landscape expansion, e.g., DoRA and PiSSA [20, 24].

**Algorithm II: SGD based expansion.** The previous implementation uses low-rank matrices for landscape expansion. We also propose a full-rank landscape expansion method by applying on-the-fly SGD to inactive blocks. Specifically, due to the compositional structure of neural networks, the gradient of the model is computed from the deep layers to the shallow layers. The strategy is to perform an SGD update on a matrix whenever its (stochastic) gradient is available, and then immediately discard the corresponding gradient after the update. We term this approach as **BREAD-SGD**. The gradient is computed on-the-fly and only the current block's gradient needs to be stored. Thus, the memory overhead is negligible.

**Variants of expansion block selection.** Based on the derivation of (7), evaluating the gradients of the expansion matrices is inexpensive for layers $\ell + 1, \ldots, L$. However, the gradient evaluation for layers $1, \ldots, \ell - 1$ is more costly, as it requires calculating $\frac{\partial z_{j+1}}{\partial z_j}$ for $j = 1, \ldots, \ell - 1$. Therefore, one computationally efficient variant of BREAD is to add expansion matrices only for layers $\ell + 1, \ldots, L$. This leads to two strategies of selecting expansion matrices:

$$J = \begin{cases} [P], & \text{for full backward} \\ \{\ell + 1, \cdots, L\}, & \text{for efficient backward.} \end{cases} \tag{6}$$

Here, $P$ denotes the total number of expansion matrices.

## 4.3 Analysis of BREAD

**Memory cost analysis.** To simplify the analysis, we consider a $D$-layer neural network where each layer consists of one matrix with dimensions $\mathbb{R}^{m \times m}$. BCD requires storing bfloat16 weight ($2Dm^2$), float32 block weight ($4m^2$), gradient ($4m^2$), and optimizer states ($8m^2$). BREAD-LoRA introduces additional cost of storing float32 LoRA parameters, gradient, and optimizer states ($32Dmr$). BREAD-SGD introduces the cost of storing one block float16 gradient ($2m^2$). Consider a setting similar to our Llama 3.1-8B experiment where $r = 4$, $D = 32$, and $m = 4096$, BREAD-LoRA and BREAD-SGD introduces additional memory cost by approximately $1.2\%$ and $2.5\%$, respectively.

**Computational cost analysis.** We now show that the additional backward cost is also cheap, since the intermediate partial derivatives used for computing the active block's gradient can be directly used for computing expansion matrices' gradients, as we have identified in (2). Specifically, the gradient of the expansion matrix $C_j$ can be expressed as

$$\frac{\partial H}{\partial C_j} = \underbrace{\frac{\partial H}{\partial z_{L+1}} \frac{\partial z_{L+1}}{\partial z_L} \cdots \frac{\partial z_{j+2}}{\partial z_{j+1}}}_{\text{Computed in (2), } \forall j \geq \ell} \frac{\partial z_{j+1}}{\partial C_j}. \tag{7}$$

Clearly, when $j \geq \ell$, computing the (stochastic) gradient of $C_j$ only requires additional computation of $\frac{\partial(z_{j+1})}{\partial C_j}$, which is cheap given the low dimensionality after low-rank factorization representation, i.e., $C_j = U_j V_j$. We empirically measure the memory and epoch training time in Table 1.

**Convergence Analysis.** We establish the sample complexity result for BREAD under common assumptions utilized for BCD analysis; see Section D.1 for the detailed assumptions, formal theorem statement and proof.

**Theorem 4.2.** *(informal) Let $\alpha$ and $\beta$ be the step size of active block and expansion matrices, respectively. Under assumptions stated in Section D.1, BREAD with deterministic gradient is a descent method with the following property*

$$H(\boldsymbol{W}_i^{t,k+1}) - H(\boldsymbol{W}_i^{t,k}) \leq -\mathcal{O}(\alpha)\|\nabla_{W_i} H(\boldsymbol{W}_i^{t,k})\|^2 - \mathcal{O}(\beta) \sum_{j \in [D] \setminus i} \|\nabla_{C_j} H(\boldsymbol{W}_j^{t,k})\|^2, \quad \tag{8}$$

*where $W_i^{t,k}$ denotes the model parameter at $k$-th step in block epoch $t$, block sub-problem $i$. $\alpha$ and $\beta$ are the step size of the active block and expansion matrices, respectively.*

By telescoping equation 8 across $k = 1, \cdots, K$ and $i = 1, \cdots, D$, and apply the step size rule in Section D.1, we can show that BREAD finds $\varepsilon-$approximate stationary point with $\mathcal{O}(\varepsilon^{-2})$ iterations.

| | Llama 3.1-8B | | | Llama 3.1-70B | | |
|---|---|---|---|---|---|---|
| Method | Peak Memory | GPU hours | GPU # | Peak Memory | GPU hours | GPU # |
| Adam | 208.2 GB | 39.2 | 8 A100 | 1260 GB+[(a)] | – | 16+ A100 |
| Galore | 40.5 GB | 31.2 | 1 A100 | – | – | – |
| LoRA | 25.0 GB | 23.6 | 1 A100 | 296.8 GB | 213.1 | 8 A100 |
| BAdam | 21.8 GB | 10.6 | 1 A100 | 276.2 GB | 119.0 | 8 A100 |
| BREAD-LoRA | 23.2 GB | 13.8 | 1 A100 | 288.6 GB | 152.7 | 8 A100 |
| BREAD-SGD | 23.5 GB | 11.2 | 1 A100 | 292.1 GB | 128.1 | 8 A100 |

Table 1: Memory footprint and one-epoch GPU hours for finetuning Llama 3.1 models on MathInstruct dataset. The BREAD's training time is based on the partial implementation introduced in (6). [(a)] Estimated memory cost.

# 5 Experiments

We evaluate the proposed BREAD in finetuning Llama 3.1-8B and Llama 3.1-70B model on math finetuning and instruction tuning tasks, comparing its memory cost, time cost and downstream performance with full training algorithm and memory efficient baselines.

## 5.1 Setup

We begin by introducing the experimental setup.

**Baselines.** We compare BREAD with **1) BAdam** [21], which applies vanilla BCD algorithm with Adam as the inner solver; **2) LoRA** [13], which freezes the pre-trained weight and only updates the injected low-rank adapters; **3) Galore** [43], which projects the gradient into low-rank spaces for reducing the memory cost; **4) Adam** [14], which serves as the full parameter training baseline.

**Instruction tuning.** We perform supervise finetuning on the Llama 3.1-8B model using Alpaca-GPT4 dataset [29], which contains 52K questions and corresponding GPT-4 generated answers. The model is evaluated on MT-bench [44] for examining the model's instruction-following capability.

**Math finetuning.** We finetune the Llama 3.1-70B and Llama 3.1-8B models on MathInstruct dataset [42] for 3 epochs, which contains 260K questions that covers wide range of fields in mathematics. The finetuned models are evaluated on 4 in-domain mathematical benchmarks, i.e., GSM8K, MATH, NumGLUE, and AQuA [3, 9, 26, 19], and 1 out-of-domain mathematical benchmarks, i.e., SimulEq [15]. The evaluations are based on 0-shot prompt and 4-shot chain-of-thought prompt, respectively. Due to the limited computational resource, we do not include the Adam's results for 70B model. Since there is no model parallel implementation released for Galore by the finish of the manuscript, we are unable to report its 70B results as well.

**Preference optimization.** After the instruction tuning, we further align the tuned model using direct preference optimization (DPO) [32] on Ultrafeedback dataset [4]. We use the model finetuned by Adam in instruction tuning phase as our base model for preference optimization.

All the experiments are run for 3 epochs. The reported scores are the best one among checkpoints at epoch 1, 2, 3. The detailed hyper-parameters are presented in Section B.

## 5.2 Memory and Time Cost Measure

In Table 1, we empirically measure the peak memory cost and one epoch's time cost for BREAD-LoRA and other baseline approaches. The GPU hour is calculated as the training time $\times$ GPU number. We set the LoRA rank to 64 to keep its number of trainable parameters (0.83 billion) close to a single block of BREAD (0.86 billion).

Evidently, both BAdam and BREAD can train 8B model within 24GB memory cost, which is feasible for a single RTX3090 GPU. All of LoRA, BAdam and BREAD can be used to finetune a 70B model with a single 8 A100-40GB node. Compared with BAdam, BREAD consumes slightly higher memory cost and training time under the efficient backward scheme.

| | GSM8K | | MATH | | NumGLUE | | SimulEq | | AQuA | | Avg. | |
|---|---|---|---|---|---|---|---|---|---|---|---|---|
| **Base model: Llama 3.1-8B** | | | | | | | | | | | | |
| **Method** | 0-shot | 4-shot | 0-shot | 4-shot | 0-shot | 4-shot | 0-shot | 4-shot | 0-shot | 4-shot | 0-shot | 4-shot |
| Base model | 17.8 | 52.5 | 8.6 | 23.2 | 25.7 | 40.6 | 12.2 | 28.8 | 19.3 | 43.7 | 16.7 | 37.8 |
| Adam | 62.3 | 64.9 | 17.4 | 22.9 | 56.4 | 56.8 | 28.6 | 33.5 | 44.9 | 52.8 | 41.9 | 46.2 |
| Galore | 46.7 | 57.2 | 16.2 | 22.9 | 42.8 | 45.0 | 28.7 | 32.3 | 47.8 | 48.4 | 36.4 | 41.2 |
| LoRA-rank80 | 48.7 | 58.1 | 13.7 | 23.0 | 34.6 | 54.4 | 29.6 | 29.0 | 47.3 | 50.3 | 34.8 | 43.0 |
| BAdam | 53.9 | 58.3 | 17.2 | 23.6 | 53.7 | 57.2 | 32.5 | 32.8 | 50.4 | 49.6 | 41.5 | 44.3 |
| **BREAD-LoRA** | 57.0 | 57.6 | 20.0 | 23.7 | 55.9 | 58.2 | 32.5 | **32.8** | 49.6 | 50.0 | 43.0 | 44.5 |
| **BREAD-LoRA+** | **57.8** | **61.8** | **20.4** | **24.6** | **56.1** | **58.8** | **32.9** | 32.7 | **51.2** | **51.0** | **43.7** | **45.8** |
| **BREAD-SGD** | 56.9 | 60.6 | 19.6 | 21.4 | 54.1 | 58.2 | 31.5 | 31.8 | 48.0 | 50.8 | 42.0 | 44.6 |
| **Base model: Llama 3.1-70B** | | | | | | | | | | | | |
| **Method** | 0-shot | 4-shot | 0-shot | 4-shot | 0-shot | 4-shot | 0-shot | 4-shot | 0-shot | 4-shot | 0-shot | 4-shot |
| Base model | 58.8 | 79.4 | 24.9 | 41.4 | 43.7 | 55.8 | 26.3 | 38.1 | 52.0 | 64.2 | 41.1 | 51.2 |
| LoRA-rank64 | 83.8 | 82.0 | **41.7** | 44.2 | 70.4 | 69.0 | 40.3 | 48.8 | 61.4 | 65.8 | 59.5 | 62.0 |
| BAdam | 81.4 | 82.9 | 40.3 | 43.8 | 68.1 | 69.7 | 50.0 | 52.7 | 65.3 | 70.1 | 61.0 | 63.8 |
| **BREAD-LoRA** | 83.4 | **84.2** | 41.4 | **44.7** | **73.1** | **74.4** | 51.3 | 56.8 | **68.3** | **70.5** | 63.5 | **66.1** |
| **BREAD-LoRA+** | **85.6** | 82.4 | 41.5 | 44.4 | 72.3 | 73.9 | 51.3 | 59.3 | 68.5 | 69.7 | **63.8** | 65.9 |
| **BREAD-SGD** | 82.8 | 83.9 | 40.8 | 43.7 | 68.9 | 69.7 | 49.7 | **61.3** | 62.2 | 69.7 | 60.9 | 65.7 |

Table 2: Math benchmark results for models finetuned on MathInstruct dataset.

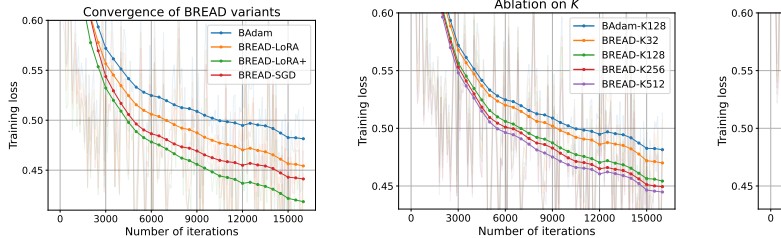
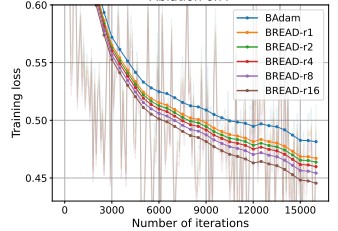

Figure 1: **(a)** Training loss of BAdam and BREAD variants; **(b)** Effect of block switch frequency $K$ for BREAD-LoRA; **(c)** Effect of the rank of expansion matrices for BREAD-LoRA.

**Remark.** The reported memory cost is higher than the theoretical value, especially for the 70B model's experiments which requires distributed training. This additional memory cost arises from storing activation values and computational buffers, e.g. the gradient buffer for performing reduce scatter operation. Furthermore, the training time may have slight fluctuations for different runs.

## 5.3 Finetuning Performance

**Math finetuning.** The evaluation results on math benchmarks are shown in Table 2. For the 8B model's finetuning, all the three BREAD variants outperform BAdam in both 0-shot and 4-shot average score. The BREAD-LoRA+ attains the highest average score, which indicates that the benefits of landscape expansion-approaches will also enhance the BREAD's performance. As for the finetuning of 70B model, the best average 0/4-shot score are obtained by BREAD-LoRA and BREAD-LoRA+, respectively.

| Method | SFT | | DPO | |
|---|---|---|---|---|
| | GPT-4 | GPT-4o | GPT-4 | GPT-4o |
| Base model | 6.07 | 4.64 | 6.63 | 5.03 |
| Adam | 6.63 | 5.03 | 7.83 | 6.08 |
| LoRA | 6.52 | 4.85 | 7.48 | 5.95 |
| Galore | 6.33 | 4.78 | 6.99 | 5.83 |
| BAdam | 6.53 | 4.88 | 7.63 | 5.99 |
| **BREAD-LoRA** | **6.77** | **5.08** | **7.68** | 6.12 |
| **BREAD-LoRA+** | 6.65 | 4.88 | 7.62 | **6.15** |
| **BREAD-SGD** | 6.68 | 4.98 | 7.52 | 5.93 |

Table 3: MT-bench scores of finetuning Llama 3.1-8B.

**Instruction tuning and DPO.** We report the MT-bench score evaluated by both GPT-4 and GPT-4o models in Table 3. After SFT, the MT-bench score of all baseline approaches improves over the base model. BREAD-LoRA achieves the highest scores in both evaluations, which are even higher than Adam, demonstrating the effectiveness of landscape expansion. Based on the model finetuned by Adam, we further align the model using direct preference optimization (DPO). Notably, BREAD-LoRA and BREAD-LoRA+ achieves the highest evaluation score by GPT-4 and GPT-4o model, respectively.

## 5.4  Convergence Verification

We present 1-epoch training loss of selected BREAD variants in Figure 1(a). For reference, we also display the loss of BAdam. One can see that all the landscape expansionapproaches accelerates BAdam, which justifies the effectiveness of landscape expansion. The BREAD-SGD is faster than BREAD-LoRA, which may attribute to the higher learning rate of the expansion matrices and the high-rank update. Notably, the BREAD-LoRA+ attains the fastest convergence, which is due to more efficient learning rate assignment of adaptors.

## 5.5  Ablation Study

In this section, we conduct ablation study to examine the effect of the expansion matrices' rank $r$ and the block switch frequency $K$ of BREAD-LoRA.

**Effect of $K$.** We present the effect of sub-problem update steps $K$ in Figure 1(b), which is by default 128 in our paper's experiments. Evidently, BREAD outperforms BAdam under all choices of $K$. Notably, increasing $K$ consistently accelerates the convergence of BREAD for the examined range, where BREAD with $K = 512$ takes only half of the iterations to reach the final training loss of BREAD with $K = 32$. One possible explanation for the phenomenon is that when using larger $K$, the Adam update will aggregate more historical information in its momentum and second moment term, which leads to better search direction and scaling magnitudes. We leave the scientific study of $K$ as a future direction.

**Effect of $r$.** The effect of expansion matrices' rank $r$ is shown in Figure 1(c), which is set to 8 in our paper. We note that by adding rank-1 expansion matrices, BREAD converges significantly faster than BAdam, which corroborates our observation in Theorem 4.1. BREAD exhibits faster convergence as the rank increases, since larger rank offers higher freedom of search directions.

## 5.6  Additional Experimental Results

We conduct additional experiments for comprehensive study of the algorithm's property. Our main findings are: **1) Convergence versus time**. In terms of wall-clock time, BREAD-SGD under efficient backward yields the fastest convergence among the BREAD variants. **2) Effect of ordering strategies.** Different block ordering yields similar convergence behavior; see Section E for the detailed results.

## 6  Conclusion and Discussions on Limitations

This paper investigates the application of a classic optimization method, known as BCD, to the finetuning of LLMs. We pinpoint two primary shortcomings of the standard BCD approach when applied to deep neural networks: the unnecessary computational overhead during backpropagation, and the misguiding optimization landscape caused by frozen blocks. To overcome these challenges, we introduce a new method termed BREAD, which unfreezes the inactive blocks and updates them in a lightweight manner. Our experimental results demonstrate that BREAD significantly enhances downstream task performance while maintaining the original BCD's memory efficiency.

**Limitations.** The convergence theory for BREAD is derived based on SGD expansion, which eases the analysis compared to LoRA-based expansion. Additionally, our analysis is based on deterministic gradient rather than stochastic setting. We leave the analysis for LoRA-based expansion and stochastic setting as future work.

**Broader impacts.** Our proposed method significantly accelerates BCD method for LLM training. This is a technical algorithmic contribution that does not yield explicit negative societal impacts. However, it carries a risk of misuse.

## Acknowledgments and Disclosure of Funding

We thank the Area Chair and the anonymous reviewers for their insightful comments and suggestions, which have significantly improved the quality of the manuscript.

Xiao Li is supported in part by the National Natural Science Foundation of China (NSFC) under grants 12201534, 12571330, and 12326608, in part by the 1+1+1 CUHK-CUHK(SZ)-GDSTC Joint Collaboration Fund under grant 2025A0505000049, and in part by the Shenzhen Science and Technology Program under grant RCYX20221008093033010.

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

# Contents

# A More Related Works

**Block coordinate descent method.** Block coordinate descent (BCD) is a classic optimization paradigm that dates back at least to [10]. It has gained popularity in recent years, due to its scalability and efficiency for many machine learning applications [27, 33, 30, 6, 31]. The community seems to converge to a consensus that, in order for BCD to be efficient, the problem it optimizes needs to possess the so-called coordinate-friendly structure [35]. Nevertheless, deep networks are of a compositional nature and not coordinate-friendly, which is perhaps why recent surveys or books have never mentioned training deep networks as an application of BCD [38, 35, 1, 39, 34]. Recently, BAdam [21] was proposed to finetune LLMs based on the BCD framework, where each block sub-problem is approximately solved using several Adam steps. Although BAdam achieved preliminary success, it is based on the vanilla BCD framework and shares the fundamental limitations we revealed in this work. In light of these, we believe identifying the limitations of BCD for LLM fintuning and fixing them entail certain insights, and this is what makes our contributions non-trivial and valuable.

**Memory efficient finetuning.** To address memory issue, multiple variants have been proposed. Parameter efficient finetuning (PEFT) methods achieve memory efficiency by only training small portion of (possibly extra) parameters while freezing most of the others, such as Adapter tuning [11], prompt tuning, and prefix tuning [16, 17]. Low-rank adaptation (LoRA) is perhaps the most popular technique that approximates model updates using two smaller, trainable low-rank matrices [13]. LoRA' variants have been proposed to address its rank constraints and further reducing the memory cost [18, 40, 5]. The work [28] proposes to use a layer-wise importance sampling for achieving memory efficiency. Galore [43] projects the gradient into low-rank space so that it does not need to store the full gradient and optimizer states in the memory. LOMO updates parameters in real time during the backpropagation process [22], so that one can perform SGD without store stochastic gradients. MeZO offers an alternative by approximating SGD using only forward passes [23], drawing from zeroth-order optimization that estimates stochastic gradients through the difference in function values. While this paper addresses the same application as these methods, they remain orthogonal to the proposed approaches. They can function as lightweight updates in the frozen layers for landscape expansion.

# B Detailed Experimental Setup

We introduce the detailed hyperparameters and experimental setup in this section.

**Global setup.** For all the experiments in math finetuning, instruction tuning and direct preference optimization, we fix the effective batch size to be 16 and train the model for 3 epochs. We use DeepSpeed ZeRO-3 to implement all the experiments that require distributed training (shown in Table 1). For all the experiments, we apply gradient checkpointing to reduce the memory cost for storing activation values. We use mixed-precision training with BFloat 16 as the low-precision datatype except for Galore, where we follow the setup in its paper, using pure BFloat 16 and 8-bit Adam optimizer for reducing the memory cost. We apply cosine learning rate schedule for all the experiments. For instruction tuning task, we set all method's initial learning rate to 1e-6. For math finetuning tasks, we set Adam's initial learning rate to 1e-6 and other methods' initial learning rate to 1e-5. The learning rate ratio for LoRA+ method is set to 16. The implementation of BAdam, Galore, LoRA are based on LLama-Factory [45].

**Math finetuning.** We randomly select 100,000 samples from the MathInstruct dataset and finetune all the models using the same samples. The benchmarks scores are evaluated using the MAmmoTH's repository[*] (without using program-of-thought). The rank of correction matrices $U$ and $V$ for BREAD-LoRA and BREAD-LoRA+ are set to 8. We initialize $U$ as zero, and initialize $V$ from the Kaiming uniform distribution [8], i.e. $\left(-\mathcal{U}(\frac{\sqrt{6}}{r}), \mathcal{U}(\frac{\sqrt{6}}{r})\right)$. The rank of LoRA is set to 80 and 64 for finetuning Llama 3.1-8B and Llama 3.1-70B, respectively, so that the trainable parameter number of LoRA is close to that of one BAdam/BREAD active block. We follow the conventional setup to set the LoRA scaling factor $\alpha = 4\times$ LoRA rank. We set Galore's rank to be 256, with the period of re-calculating the projection matrix being 256. We set $K = 100$ for BAdam and BREAD.

---

[*]https://github.com/TIGER-AI-Lab/MAmmoTH

**Instruction tuning and direct preference optimization.** The evaluation of MT-bench score is based on FastChat [44] using both GPT-4 and GPT-4o. We use the API "gpt-4-32k_0613" and "gpt-4o_2024-11-20" for GPT-4 and GPT-4o, respectively. The maximum sequence is set to 1024 and 2048 for the experiments on Alpaca-GPT4 and UltraFeedback, respectively. For BAdam and BREAD, we solve each block sub-problem for 128 steps, i.e. $K = 128$.

## C  BREAD-SGD Algorithm

We introduce a variant of the BREAD algorithm, termed BREAD-SGD, which employs Adam for updating the active block and on-the-fly SGD for the inactive block. The detailed procedure is outlined in Algorithm 2. Analogous to BREAD, BREAD-SGD partitions the model into $D$ distinct blocks and combines the gradient computation and update steps into a singular operation. Specifically, gradients are calculated on a layer-by-layer basis; active layers are updated using Adam, while inactive layers undergo a single SGD step. Once an inactive layer is updated, its gradient is discarded to enhance memory efficiency.

---

**Algorithm 2** BREAD (SGD variant)

---

1: **Input:** Model parameters $\{\boldsymbol{W}_\ell^0\}_{\ell=1}^L$, number of blocks $D$, iterations per block $K$, step size of inactive blocks $\beta$
2: **Initialization:** Block-epoch index $t \leftarrow 0$, and the corresponding optimizer states $\tilde{\boldsymbol{s}}_j^0 \leftarrow \boldsymbol{0}$, $\forall j \in [P]$
3: **while** stopping criterion not met **do**
4:      Generate a block partition $\pi = \{\pi_1, \ldots, \pi_D\}$
5:      **for** one *block-epoch* $i = 1$ to $D$ **do**
6:          Select correction matrices' indices $J \subset [P]$ as in (6)
7:          $\boldsymbol{s}_{\pi_i}^{t,0} \leftarrow \boldsymbol{0}$    // Re-initialize Adam optimizer states
8:          $\boldsymbol{W}_{\pi_i}^{t,0} \leftarrow \boldsymbol{W}_{\pi_i}^t$
9:          **for** *landscape corrected block updates* $k = 1$ to $K$ **do**
10:             Sample a data batch in random reshuffled order $\mathcal{D}_B = \{(x_j, y_j)\}_{j=1}^B \sim \mathcal{D}$
11:             **within** one backward pass on the data batch $\mathcal{D}_B$ **do**
12:                // Update the active block
13:                $\boldsymbol{g}_{\pi_i}^{t,k} \leftarrow \frac{\partial H}{\partial \boldsymbol{W}_{\pi_i}}$
14:                $\boldsymbol{W}_{\pi_i}^{t,k}, \boldsymbol{s}_{\pi_i}^{t,k} \leftarrow \mathsf{AdamStep}(\boldsymbol{W}_{\pi_i}^{t,k-1}, \boldsymbol{g}_{\pi_i}^{t,k}, \boldsymbol{s}_{\pi_i}^{t,k-1})$
15:
16:                // Correct inactive blocks
17:                **for** $\ell \in J$ **do**
18:                    $\boldsymbol{g}_\ell^{t,k-1} \leftarrow \frac{\partial H}{\partial \boldsymbol{W}_\ell}$
19:                    $\boldsymbol{W}_\ell^{t,k} \leftarrow \boldsymbol{W}_\ell^{t,k-1} - \beta \boldsymbol{g}_\ell^{t,k-1}$
20:                    $\boldsymbol{g}_\ell^{t,k-1} \leftarrow$ **None**
21:                **end for**
22:             **end within**
23:          **end for**
24:          $\boldsymbol{W}_{\pi_i}^{t+1} \leftarrow \boldsymbol{W}_{\pi_i}^{t,K}; \boldsymbol{s}_{\pi_i}^{t,K} \leftarrow$ **None**
25:      **end for**
26:      $t \leftarrow t + 1$
27: **end while**
28: Return parameters $\{\boldsymbol{W}_\ell^t\}_{\ell=1}^L$

---

# D Convergence Result and Additional Proofs

## D.1 Convergence of BREAD

In this section, we establish a preliminary convergence result for the proposed algorithms. For conciseness, we analyze the BREAD-SGD described in Algorithm 2. The analysis of BREAD and BREAD-partial follows a similar approach.

Our analysis decouples the error terms induced by the update of active block and inactive blocks, respectively. The descent property of the algorithm is established by analyzing consecutive updates across one block epoch. We follow the deterministic setting in [21], leaving the analysis of stochastic cases for the future work. We now introduce the assumptions used in analysis.

**Assumption D.1** (Smoothness)**.** The function $H(\boldsymbol{W})$ is $L$-smooth on $\boldsymbol{W}$ and $L_i$-smooth on block $\boldsymbol{W}_i$ for blocks $i = 1, \cdots, D$:

$$\|\nabla_{\boldsymbol{W}} H(\boldsymbol{W}) - \nabla_{\boldsymbol{W}} H(\bar{\boldsymbol{W}})\| \leq L\|\boldsymbol{W} - \bar{\boldsymbol{W}}\|. \tag{9}$$

$$\left\| \frac{\partial H}{\partial \boldsymbol{W}_i}\bigg|_{\boldsymbol{W}_i^1} - \frac{\partial H}{\partial \boldsymbol{W}_i}\bigg|_{\boldsymbol{W}_i^2} \right\| \leq L_i \|\boldsymbol{W}_i^1 - \boldsymbol{W}_i^2\|, i = 1, \cdots, D. \tag{10}$$

We denote $\bar{L} = \max_{i \in [D]} L_i$ as the maximum Lipschitz constant.

**Assumption D.2** (Bounded derivatives)**.** The block derivatives $\nabla_{\boldsymbol{W}_i} H(\boldsymbol{W})$ are uniformly bounded by a constant $G$ along the update trajectory of BREAD:

$$\|\nabla_{\boldsymbol{W}_i} H(\boldsymbol{W})\| \leq G, i = 1, \cdots, D.$$

The Theorem D.1 is standard for analyzing block coordinate descent-type methods [38]. Note that the block-wise smoothness (10) can be naturally induced by the function smoothness (9). The Theorem D.2 is provably satisfied for Adam algorithm under certain generalized smoothness conditions. The Theorem D.2 directly induces the following corollary, which will be used in our analysis.

**Corollary D.3** ([21], Corollary D.3)**.** *Let* $A_i^{t,k} := diag\left(1/\left(\sqrt{\hat{v}_i^{t,k}} + \lambda\right)\right)$ *denote a diagonal matrix formed by coordinate-wise adaptive step sizes vector. Under Theorem D.2 and with $0 < \lambda < 1$, we have*

$$\frac{1}{2G}I \preccurlyeq A_i^{t,k} \preccurlyeq \frac{1}{\lambda}I. \tag{11}$$

**Lemma D.4** (Re-statement of [21], Lemma D.7)**.** *Under the Theorem D.1 and Theorem D.2, and given the step size condition $\alpha \leq \frac{\lambda}{2\bar{L}K^2}$, we have the following bound for the bias between Adam update and GD update $\|e_i^t\| := \frac{1}{K}\sum_{k=1}^K \frac{1}{1-\beta^k}\|H_i^{t,k}(m_i^{t,k} - (1-\beta_1^k)g_i^{t,1})\|$ :*

$$\|e_i^t\|_2^2 \leq \frac{2L_i\alpha K}{\lambda^2}\|g_i^{t,1}\|_2^2.$$

**Theorem D.5** (Descent of BREAD)**.** *Under Theorem D.1 and Theorem D.2, the Algorithm 2 with deterministic gradient achieves the following descent after each block-epoch of updates:*

$$H(\boldsymbol{W}^{t,k+1}) - H(\boldsymbol{W}^{t,k}) \leq -\mathcal{O}(\alpha)\|\nabla_i H(\boldsymbol{W}^{t,k})\|^2 - \mathcal{O}(\beta)\|\nabla_j H(\boldsymbol{W}^{t,k})\|^2, \tag{12}$$

*under the step size choice $\beta = \frac{2\alpha}{G}$, $\alpha \leq \min\{\frac{\lambda}{2\bar{L}K^2}, \frac{\lambda^2}{24\bar{L}KG}, \frac{G}{4\bar{L}K}, \sqrt{\frac{6G^2}{16\bar{L}K}}\}$.*

By telescoping (12) from $t = 1, \cdots, T$, and divide each side by $T$, we obtain the sample complexity of $\|\nabla H(\boldsymbol{W}^t)\|^2 = \mathcal{O}(K/T)$. The proof of Theorem D.5 is based on the following Lemma, which establishes the descent property of one block sub-problem.

**Lemma D.6** (Descent of one block-epoch)**.** *Under Theorem D.1 and Theorem D.2, the Algorithm 2 update yields the following approximate descent property:*

$$H(\boldsymbol{W}_i^t) - H(\boldsymbol{W}_{i-1}^t) \leq -\frac{\alpha K}{4G}\|\nabla H(\boldsymbol{W}_{i-1}^t)\|_2^2$$

*Proof.* The proof of Theorem D.6 is based on the analysis framework of [21]. Additionally, we need to analyze the descent property of the update in correction matrices. At the beginning of block epoch $t$, block sub-problem $i$, let $\boldsymbol{W}_{j,i}^{t}$ be the parameter of block $j$, $\boldsymbol{W}_{i}^{t}$ be the full parameter. Let $\beta$ be the step size for the correction matrices. Based on the smoothness property in Theorem D.1, we have

$$H(\boldsymbol{W}_i^{t,k+1}) - H(\boldsymbol{W}_i^{t,k})$$

$$\leq \sum_{j\in[D]\setminus i} \left\langle \nabla_j H(\boldsymbol{W}_i^{t,k}), \boldsymbol{W}_{j,i}^{t,k+1} - \boldsymbol{W}_{j,i}^{t,k} \right\rangle + \sum_{j\in[D]\setminus i} \frac{L}{2}\|\boldsymbol{W}_{j,i}^{t,k+1} - \boldsymbol{W}_{j,i}^{t,k}\|_2^2$$

$$+ \left\langle \nabla_i H(\boldsymbol{W}_i^{t,k}), \boldsymbol{W}_{i,i}^{t,k+1} - \boldsymbol{W}_{i,i}^{t,k} \right\rangle + \frac{L}{2}\|\boldsymbol{W}_{i,i}^{t,k+1} - \boldsymbol{W}_{i,i}^{t,k}\|_2^2$$

$$= \sum_{j\in[D]\setminus i} (-\beta + \frac{L}{2}\beta^2)\|\nabla_j H(\boldsymbol{W}_i^{t,k})\|_2^2 + \left\langle \nabla_i H(\boldsymbol{W}_i^{t,k}), \boldsymbol{W}_{i,i}^{t,k+1} - \boldsymbol{W}_{i,i}^{t,k} \right\rangle + \frac{L}{2}\|\boldsymbol{W}_{i,i}^{t,k+1} - \boldsymbol{W}_{i,i}^{t,k}\|_2^2$$

$$\leq \sum_{j\in[D]\setminus i} (-\beta + \frac{L}{2}\beta^2)\|\nabla_j H(\boldsymbol{W}_i^{t,k})\|_2^2 + (-\alpha + \frac{L}{2}\alpha^2)\|\nabla_i H(W_i^{t,k})\|_2^2 + \alpha\left\langle \nabla_i H(\boldsymbol{W}_i^{t,k}), e_i^{t,k} \right\rangle$$

$$+ \frac{L_i}{2}(\alpha G + L_i\alpha^2 k^2)\|e_i^{t,k}\|^2, \tag{13}$$

where the equalities is due to the GD update rule, and the last inequality uses Young's inequality and the Adam's update rule. $\|e_i^{t,k}\| := \frac{1}{K}\sum_{k=1}^{K} \frac{1}{1-\beta_1^k}\|H_i^{t,k}(m_i^{t,k} - (1-\beta_1^k)g_i^{t,k})\|$ is the bias of active block's update, where we use $g_i^{t,k}$ to represent $\nabla_i H(W_{i,i}^{t,k})$ for brevity. We have

$$\|e_i^{t,k}\| = \frac{1}{k}\sum_{n=1}^{k} \frac{1}{1-\beta_1^n}\|H_i^{t,n}(m_i^{t,n} - (1-\beta_1^n)g_i^{t,n})\|$$

$$\leq \frac{1}{k}\sum_{n=1}^{k} \frac{1}{(1-\beta_1^n)\lambda}\|\sum_{z=1}^{n} \beta_1^{n-z}\left(g_i^{t,n} - g_i^{t,k}\right)\|$$

$$\leq \frac{1}{k}\sum_{n=1}^{k} \frac{1}{(1-\beta_1^n)\lambda}\sum_{z=1}^{n} \beta_1^{n-z}L_i\|W_{i,i}^{t,z} - W_{i,i}^{t,k}\|. \tag{14}$$

For the ease of expression, let $\theta_i^{t,k} := W_{i,i}^{t,k}$. Define $\Delta_i^t(k,z) := \|\theta_i^{t,k} - \theta_i^{t,z}\|$. We have

$$\Delta_i^t(k,z) = \alpha\|\sum_{j=1}^{z} H_{i,i}^{t,j}m_{i,i}^{t,j}\|$$

$$\leq \alpha\sum_{j=1}^{z} \frac{1}{\lambda(1-\beta_1^j)}\|(1-\beta_1)\sum_{l=1}^{j} \beta_1^{j-l}g_i^{t,l}\|$$

$$= \alpha\sum_{j=1}^{z} \frac{1}{\lambda(1-\beta_1^j)}\|(1-\beta_1)\sum_{l=1}^{j} \beta_1^{j-l}(g_i^{t,l} - g_i^{t,z}) + (1-\beta_1^k)g_i^{t,z}\|$$

$$\leq \alpha\sum_{j=1}^{z} \frac{1}{\lambda(1-\beta_1^j)}(1-\beta_1)\sum_{l=1}^{j} \beta_1^{j-l}L_i\|\theta_i^{t,l} - \theta_i^{t,z}\| + (1-\beta_1^k)\|g_i^{t,z}\|$$

$$\leq \alpha\sum_{j=1}^{z} \frac{1}{\lambda(1-\beta_1^j)}(1-\beta_1)\left(\sum_{l=1}^{j} \beta_1^{j-l}L_i\Delta_i^t(k,z)\sum_{l=1}^{j} \beta_1^{j-l} + (1-\beta^k)g_i^{t,z}\right)$$

$$\leq \frac{\alpha z^2}{\lambda}\left(L_i\Delta_i^t(k,z) + \|g_i^{t,z}\|\right) \tag{15}$$

Combine (15), (14) and apply the step size rule, we have

$$\|e_i^{t,k}\| \leq \frac{2L_i\alpha}{\lambda^2}\|g_i^{t,k}\|. \tag{16}$$

Plug (16) back into (13), we have

$$H(\boldsymbol{W}_i^{t,k+1}) - H(\boldsymbol{W}_i^{t,k}) \tag{17}$$

$$\leq \sum_{j\in[D]\setminus i} (-\beta + \frac{L_j}{2}\beta^2)\|\nabla_j H(\boldsymbol{W}_i^{t,k})\|_2^2 + (-\alpha + \frac{L_i}{2}\alpha^2)\|\nabla_i H(W_i^{t,k})\|_2^2$$

$$- \left(\frac{8L_i^2\alpha^2 k^2 G^2}{\lambda^4} + \frac{8L_i^3\alpha^3 k^3 G}{\lambda^4}\right)\|\nabla_i H(W_i^{t,k})\|_2^2 \tag{18}$$

Based on Theorem D.4 and use the fact that $\|\nabla_i H(W_{i-1})\|_2 \leq \|\nabla H(W_{i-1})\|_2$, we can derive the following bound:

$$H(\boldsymbol{W}_i^{t,k+1}) - H(\boldsymbol{W}_i^{t,k}) \leq -\frac{\alpha k}{8G}\|\nabla_i H(W_i^{t,k})\|_2^2 - \sum_{j\in[D]\setminus i} -\frac{\beta k}{8G}\|\nabla_j H(W_i^{t,k})\|_2^2. \tag{19}$$

*Proof of Theorem D.5.* (19) establishes exact the same descent property as in [21] Corollary D.8. By following the same argument as in [21] (10)–(11), one can establish the convergence result in Theorem D.5.

$\square$

## D.2 Proof of Propositions

*Proof of Theorem 3.1.* We first show that $H^* = 0$. One can construct $\boldsymbol{z}_3 = [1, 0, \cdots, 0]^\top$ and $\boldsymbol{W}_3 = [\boldsymbol{y}, \boldsymbol{0}, \cdots, \boldsymbol{0}]$. Note that such a choice of $\boldsymbol{z}_3$ is always achievable by choosing a specific $\boldsymbol{W}_2$. Hence, 0 function value can be attained by the constructed feasible point. This yields $H^* = 0$ after realizing that the objective function must be nonnegative.

In BCD, $\boldsymbol{W}_1$ and $\boldsymbol{W}_3$ are fixed. We further assume that the fixed $\boldsymbol{W}_3$ has full column rank. We split our discussion into two cases. Case I: $\boldsymbol{y} \notin \text{range}(\boldsymbol{W}_3)$. We trivially have $\widetilde{H}^* > 0 = H^*$. Case II: $\boldsymbol{y} \in \text{range}(\boldsymbol{W}_3)$. In this case, $\boldsymbol{z}_3^* := (\boldsymbol{W}_3^\top \boldsymbol{W}_3)^{-1}\boldsymbol{W}_3^\top \boldsymbol{y}$ is the unique point that can achieve 0 function value. However, since $\boldsymbol{z}_3^*$ has at least one negative entry and $\boldsymbol{z}_3 \geq 0$ (due to the ReLU activation), we have $\|\boldsymbol{z}_3 - \boldsymbol{z}_3^*\|_2^2 > 0$. Therefore, we have $\|\boldsymbol{y} - \hat{\boldsymbol{y}}\|_2^2 = \|\boldsymbol{W}_3(\boldsymbol{z}_3^* - \boldsymbol{z}_3)\|_2^2 > 0 = H^*$, where the last inequality follows from the full column rankness of $\boldsymbol{W}_3$. $\square$

*Proof of Theorem 4.1.* We construct $\boldsymbol{z}_3 = \boldsymbol{e}_1 = [1, 0, \cdots, 0]^\top$. Let $\boldsymbol{C} = \left[\boldsymbol{y} - \boldsymbol{W}_3^{(1)}, 0, \cdots, 0\right]$, where $\boldsymbol{W}_3^{(1)}$ is the first column of $\boldsymbol{W}_3$. Then, we have $\|(\boldsymbol{W}_3 + \boldsymbol{C})\boldsymbol{z}_3 - \boldsymbol{y}\|_2^2 = \|\boldsymbol{C}\boldsymbol{e}_1 - (\boldsymbol{y} - \boldsymbol{W}_3\boldsymbol{e}_1)\|_2^2 = 0$. $\square$

## D.3 Analysis of Multi-Layer Model and Cross Entropy Loss

In this section, we generalize the Theorem 3.1 to $L$-layer neural network model and cross entropy loss. The corresponding numerical verification are presented in Section E. Let us consider an $L$-layer model:

$$\boldsymbol{z}_1 = \sigma(\boldsymbol{W}_1 x)$$
$$\boldsymbol{z}_i = \sigma(\boldsymbol{W}_i \boldsymbol{z}_{i-1}), \quad i = 2, \cdots, L-1$$
$$\hat{\boldsymbol{y}} = \boldsymbol{W}_L \boldsymbol{z}_{L-1},$$

where $\sigma(\boldsymbol{x}) = \max(0, \boldsymbol{x})$ is the ReLU activation function and $\boldsymbol{z}_i \in \mathbb{R}^{d_i}$.

### D.3.1 Suboptimality Analysis for $L$-layer Model

Let us first consider the general regression loss $\|\hat{\boldsymbol{y}} - \boldsymbol{y}\|_2^2$, where $\boldsymbol{y}$ is the target we aim to fit.

**Effect of freezing $\boldsymbol{W}_L$.** When $\boldsymbol{W}_L$ is full column rank, the optimal $\boldsymbol{z}_{L-1}$ we seek to fit is the least square solution $\boldsymbol{z}_{L-1}^* = (\boldsymbol{W}_L^\top \boldsymbol{W}_L)^{-1}\boldsymbol{W}_L^\top \boldsymbol{y}$. When $\boldsymbol{z}_{L-1}^*$ contains negative entries, it cannot be fit due to the non-negativity of the ReLU function, which induces the suboptimality:

$$\min_{\boldsymbol{W}_1, \cdots \boldsymbol{W}_{L-1}} \|\hat{\boldsymbol{y}} - \boldsymbol{y}\|_2^2 > \min_{\boldsymbol{W}_1, \cdots, \boldsymbol{W}_L} \|\hat{\boldsymbol{y}} - \boldsymbol{y}\|_2^2.$$

**Effect of freezing intermediate layers.** Each intermediate layer performs the transformation $z_i = \sigma(W_i z_{i-1}) := \mathcal{M}_i$. When $W_i$ is trainable, we have range$(\mathcal{M}_i) = \mathbb{R}^{d_i+}$ when $z_{i-1} \neq 0$. However, when $W_i$ is frozen, range$(\mathcal{M}_i)$ is limited to the projected "restricted" column space of $W_i$, where "restricted" means that the column combination should be positive, due to the positivity of $z_{i-1}$.

### D.3.2 Suboptimality analysis for cross entropy loss

Without loss of generality, let us assume that the ground truth label is the first class. The cross entropy loss is given by $-\log\left(\exp \hat{y}_1 / (\sum_{i=1}^{m} \exp \hat{y}_i)\right)$, where $m$ is the number of classes. Consider the case where the weight of the last layer $W_L$ has a row with the same weight as the first row, i.e. $\exists\, j$ such that $W_L^{(j)} = W_L^{(1)}$, we have $\hat{y}_j = W_L^{(j)} z_{L-1} = W_L^{(1)} z_{L-1} = \hat{y}_1$. In this case, we will never be able to drive the loss down to $-\log \frac{1}{2}$:

$$-\log\left(\frac{\exp \hat{y}_1}{(\sum_{i=1}^{m} \exp \hat{y}_i)}\right) > -\log\left(\frac{\exp \hat{y}_1}{\exp \hat{y}_1 + \exp \hat{y}_i}\right) = -\log \frac{1}{2}.$$

While it is not common for $W_L$ to have exactly two same rows, one can expect large error when there are rows that form small angle, i.e.

$$\frac{\left\langle w_L^{(i)}, w_L^{(j)} \right\rangle}{\|w_L^{(i)}\| \|w_L^{(j)}\|} \approx 1.$$

### D.3.3 Small Experiment on Multi-layer Neural Network Training

Below, we conduct a small experiments on training multi-layer neural network to demonstrate the aforementioned suboptimality issue. We treat each layer as one block, and set the block switch frequency $K = 100$. As shown in Figure 2(a), BREAD-LoRA with rank-1 landscape correction converges dramatically faster than BAdam. In particular, BAdam only begins to converge rapidly after the 300 steps, when the final layer has been trained. This phenomenon supports our discussion that a poorly trained final layer may hinder convergence. In Figure 2(b), we show that BREAD boosts the convergence for 8-layer neural network as well, which corroborates our $L$-layer analysis in Section D.3.1.

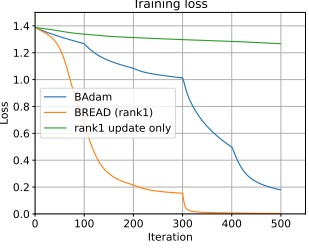
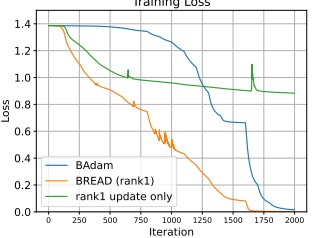

(a) Training loss of 3-layer model.  (b) Training loss of 8-layer model.

Figure 2: BREAD with rank-1 landscape expansion converges dramatically faster than BCD.

# E   Additional Experiments

**Convergence in time.** We finetune the Llama 3.1-8B model on MathInstruct dataset for 3 epochs, and report training loss convergence versus time/iteration in Figure 3. Notably, BREAD-SGD-partial achieves the fastest convergence in terms of time, and BREAD-SGD and BREAD-partial surpasses BAdam at certain points. All the BREAD variants achieve lower training loss than BAdam after 3 epochs.

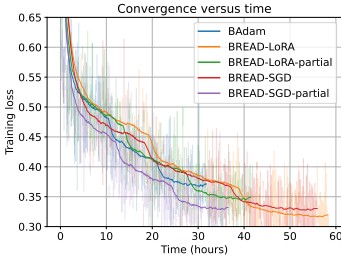 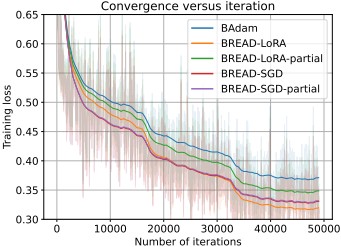

(a) Training loss in terms of time.   (b) Training loss in terms of iteration.

Figure 3: Convergence analysis of training loss in terms of time and iteration.

**Effect of ordering strategies.** We test the ordering strategies of ascending (from input layer to output layer), descending (from output layer to input layer), and random (select the layer in random reshuffling manner). As shown in Figure 4, different ordering strategy does not result in evident difference of convergence speed.

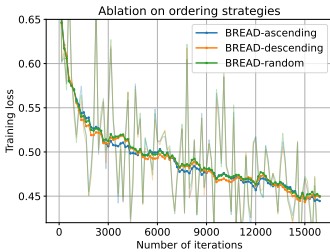

Figure 4: Ablation study on block ordering strategies

