# OpenReview forum: "Accelerating Block Coordinate Descent for LLM Finetuning via Landscape Expansion"
_NeurIPS.cc/2025/Conference — NeurIPS 2025 poster_

### Official Review · Reviewer_GYiG · 2025-06-28

**Clarity:** 3
**Significance:** 2
**Originality:** 3
**Rating:** 5
**Confidence:** 3

**Summary:**

This paper introduces BREAD, a memory-efficient algorithm for fine-tuning LLMs that improves upon Block Coordinate Descent (BCD).

While BCD reduces memory requirements by updating only one block of parameters at a time, the authors identify two critical limitations when applied to neural networks: wasted computation during backpropagation through frozen layers, and a restricted optimization landscape that can slow convergence.

BREAD addresses both issues by unfreezing inactive blocks and updating them using lightweight methods (low-rank matrices or SGD) during the same backpropagation pass. This "landscape expansion" adds minimal memory overhead while utilizing already-computed gradients and providing a broader optimization view.

**Questions:**

Why choose LoRA-rank80 and LoRA-rank64 when comparing with LoRA?

suggestions:

1.For the assumptions made during the theoretical derivation, additional statistical evidence can be provided to demonstrate that these assumptions are commonly present in practice.

2.Include experimental comparisons with at least 2–3 recent memory-efficient methods.

**Ethical Concerns:**

["NO or VERY MINOR ethics concerns only"]

**Limitations:**

yes, authors addressed the limitations and potential negative societal impact of their work

**Quality:**

2

**Strengths And Weaknesses:**

Strengths：
- The theoretical analysis is thorough and compelling.The paper formally proves BCD's suboptimality issue in neural networks through Proposition 3.1 and demonstrates that rank-1 expansion can resolve it via Proposition 4.1.
- Systematic ablation experiments on key hyperparameters enhance the credibility of the method.

Weaknesses：
- Missing comparisons with some recent memory-efficient methods like ReLoRA, LISA, etc.
- Some theoretical assumptions have not been substantiated to generally hold in the fine-tuning of large language models.

---

> ### Author Rebuttal · Authors · 2025-07-30
>
> We thank the reviewer for the valuable comments. We now address your concerns below.
>
> **Missing comparison with recent memoy-efficient methods.** Thank you for the suggestion. We finetune Llama 3.1-8B on MathInstruct dataset under the same setting in Table 2 of the manuscript, using the additional ReLoRA and DoRA. The full benchmark score is shown in the following table:
>
> | Method          | GSM8K | MATH | NumGLUE | SimulEq | AQuA | Avg.  |
> |-----------------|--------|------|---------|---------|------|--------|
> | Base model      | 17.8   | 8.6  | 25.7    | 12.2    | 19.3 | 16.7   |
> | Adam            | 62.3   | 17.4 | 56.4    | 28.6    | 44.9 | 41.9   |
> | Galore          | 46.7   | 16.2 | 42.8    | 28.7    | 47.8 | 36.4   |
> | LoRA-rank80     | 48.7   | 13.7 | 34.6    | 29.6    | 47.3 | 34.8   |
> | DoRA       | 50.5   | 15.0 | 40.2    | 30.0    | 46.1 | 36.4   |
> | ReLoRA          | 47.9   | 18.3 | 49.5    | 31.1    | 45.3 | 38.4   |
> | BAdam           | 53.9   | 17.2 | 53.7    | 32.5    | 50.4 | 41.5   |
> | BREAD-LoRA      | 57.0   | 20.0 | 55.9    | 32.5    | 49.6 | 43.0   |
> | BREAD-LoRA+     | 57.8   | 20.4 | 56.1    | 32.9    | 51.2 | 43.7   |
> | BREAD-SGD       | 56.9   | 19.6 | 54.1    | 31.5    | 48.0 | 42.0   |
> **Table:** 0-shot score of math finetuning.
>
> | Method          | GSM8K | MATH | NumGLUE | SimulEq | AQuA | Avg.  |
> |-----------------|--------|------|---------|---------|------|--------|
> | Base model      | 52.5   | 23.2 | 40.6    | 28.8    | 43.7 | 37.8   |
> | Adam            | 64.9   | 22.9 | 56.8    | 33.5    | 52.8 | 46.2   |
> | Galore          | 57.2   | 22.9 | 45.0    | 32.3    | 48.4 | 41.2   |
> | LoRA-rank80     | 58.1   | 23.0 | 54.4    | 29.0    | 50.3 | 43.0   |
> | DoRA        | 57.2   | 23.1 | 52.5    | 30.9    | 47.6 | 42.3   |
> | ReLoRA          | 55.7   | 24.2 | 56.0    | 31.5    | 48.7 | 43.2   |
> | BAdam           | 58.3   | 23.6 | 57.2    | 32.8    | 49.6 | 44.3   |
> | BREAD-LoRA      | 57.6   | 23.7 | 58.2    | 32.8    | 50.0 | 44.5   |
> | BREAD-LoRA+     | 61.8   | 24.6 | 58.8    | 32.7    | 51.0 | 45.8   |
> | BREAD-SGD       | 60.6   | 21.4 | 58.2    | 31.8    | 50.8 | 44.6   |
> **Table:** 4-shot score of math finetuning.
>
> Notably, both BAdam and BREAD outperform ReLoRA and DoRA. This suggests that BCD-based algorithm might be more suitable for math finetuning task. Due to limited computational resource, we are unable to finish the experiment of LISA during the short rebuttal period. We will provide a more comprehensive comparison with memory-efficient optimization algorithms in the next version of our manuscript.
>
> **Some assumptions have not been substantiated to generally hold for LLM finetuning.** We acknowledge that the assumptions such as smoothness and deterministic gradient might not hold for general LLM training. However, without assumptions such as smoothness, it is challenging to establish a meaningful convergence result. Theoretical works (e.g., [1, 2]) that analyze Adam's convergence also require certain type of smoothness to derive the descent of the objective for each step. Additionally, while it is possible to remove the deterministic assumption, the analysis will be unnecessarily intricate and does not bring additional insight about the algorithm. The current convergence result based on deterministic gradient can already serve a partial explanatory purpose to show that the algorithm is grounded by classic convergence theory. We remark that it is possible to combine the current analysis with [1, 2] to derive the convergence under stochastic gradient. The important steps would be establishing the approximate descent property (rather descent property) using existing analysis for Adam under stochastic setting, and then providing an overall approximate descent for the BCD loop. Finally, standard telescoping will yield the complexity result. We leave the detailed analysis as future work.
>
> **Why choosing LoRA-rank80 and LoRA-rank64?** We set LoRA's rank to be 80 and 64 for finetuning 8B and 70B model, respectively. As we explained in Appendix B, we adopt the rank so that the number of trainable parameters of LoRA roughly matches the active block of BAdam/BREAD for a fair comparison.
>
> **Suggestions:**
>
> 1. **Statistical evidence for verifying assumptions.** Thank you for the suggestion. In the next version, we would like to exam whether the smoothness condition and gradient boundedness condition will be satisfied for a simple and small LLM. Note that verifying smoothness often needs to bound the spectral norm of Hessian, which is computationally prohibitive for large-scale LLMs.
>
> 2. **Include 2–3 recent memory-efficient methods.** Thank you for the suggestion. As we replied previously, we have added comparisons with ReLoRA and DoRA. We will add them in the next version of the manuscript.
>
> We hope that our response is satisfactory and addresses all your concerns. If there are any additional questions and concerns, please feel free to let us know during the author-reviewer discussion period. We are more than happy to provide further clarifications.
>
> **References**
>
> [1] Zhang et al, "Adam can converge without any modification on update rules", NeurIPS 2022.
>
> [2] Li et al, "Convergence of Adam under relaxed assumptions", NeurIPS 2023.

---

### Official Review · Reviewer_FNMe · 2025-07-01

**Clarity:** 3
**Significance:** 2
**Originality:** 2
**Rating:** 5
**Confidence:** 4

**Summary:**

Based on the recent memory-efficient fine-tuning method like BAdam, the authors address two key limitations of the BAdam optimizer: computational waste in shallow layers and a slower convergence rate.  They propose BREAD, a novel approach that combines BAdam with either LoRA or in-place SGD (IP-SGD). The paper presents theoretical analyses and several propositions to support the proposed method. The authors conduct a series of experiments under various scenarios to demonstrate the effectiveness of their approach.

**Questions:**

Besides the above things mentioned in the weakness, I have a few questions and suggestions here.

Questions:
1. A key motivation for BREAD-LoRA is to reduce wasted computation. However, a potential inefficiency appears to remain. During backpropagation in the shallow layers, are the partial derivatives for the frozen, pre-trained base model still computed? If so, this would reintroduce the same computational waste that BREAD was designed to eliminate in BAdam.
2. Since both LoRA and SGD have low memory and training overhead, it seems natural to consider combining them. How would BREAD-LoRA-SGD perform? Could it yield better results?
3. The authors propose two variants but do not clarify when one should be preferred over the other. Based on the empirical results, BREAD-LORA appears to consistently outperform the alternative.

Suggestions:
1. In line 196, it is good to acknowledge and mention the work LOMO [1] here.
2. In Algorithm 1, the notation can be improved and the author should define the notation. Ex. $s_{J}^{t,0}$ and $s_{J}^t$, and the author should also use other symbols than $J$ in line 6 to avoid confusion with $j$.
3. Since the BREAD is a hybrid method with BCD and LoRA/IP-SGD, it is also good to add the discussion of Addax [2] in their related work, which is also a memory efficient hybrid method.

I hope the authors can address these concerns satisfactorily, in which case I would consider raising my score.

[1] Lv, Kai, et al. "Full parameter fine-tuning for large language models with limited resources." arXiv preprint arXiv:2306.09782 (2023).

[2] Li, Zeman, et al. "Addax: Utilizing Zeroth-Order Gradients to Improve Memory Efficiency and Performance of SGD for Fine-Tuning Language Models." ICLR 2025.

**Ethical Concerns:**

["NO or VERY MINOR ethics concerns only"]

**Final Justification:**

Good paper. Clear accept.

**Limitations:**

yes

**Paper Formatting Concerns:**

No major formatting issues.

**Quality:**

3

**Strengths And Weaknesses:**

Strengths:

1. The motivation is clear. The paper focus on two key limitations of the BCD approach: the computational waste from calculating partial derivatives in shallow layers and the challenges of a narrow optimization landscape.

2. The proposed BREAD method offers a novel and clear approach to addressing the limitations of BAdam

3. The experimental evaluation is comprehensive, replicating the full suite of experiments from the original BAdam paper.

Weaknesses:

1. The theoretical convergence rate of  $O(\epsilon^{-2})$ (Theorem 4.2) is expected, given that the method combines Block Coordinate Descent (BCD) and SGD. The analysis could be strengthened by extending it to the fully stochastic case, a limitation the authors acknowledge.

2. A key motivation for using LoRA is parameter efficiency, as one only needs to store the adapter instead of the full pre-trained model. The proposed BREAD-LoRA method appears to forfeit this advantage by requiring the full model to be stored. Also the inference time will increase with the LoRA adapter.

3. Given that the authors propose two variants, BREAD-LoRA and BREAD-SGD, a natural extension to consider is a combined BREAD-LoRA-SGD approach. Since both LoRA and SGD individually have low memory and computational overhead, this hybrid method could potentially yield better results.

---

> ### Author Rebuttal · Authors · 2025-07-31
>
> We thank the reviewer for the valuable comments. We now address your concerns below.
>
> **The convergence analysis is based on the deterministic assumption.** Since our main focus for this work is to design a practical algorithmic framework (and its practical implementation) rather than establishing new theoretical convergence result, we consider the convergence under the deterministic setting for compact presentation. We remark that it is possible to combine the current analysis with [1, 2] to derive the convergence under stochastic gradient. The important steps would be establishing the approximate descent property (rather descent property) using existing analysis for Adam under stochastic setting, and then providing an overall approximate descent for the BCD loop. Finally, standard telescoping will yield the complexity result. We leave the detailed analysis as future work
>
> **BREAD-LoRA requires storing the full model checkpoint.** Storing the full parameter checkpoint appears to be inevitable when using full parameter optimization methods such as Adam, SGD, and BCD-based methods. While LoRA has the advantage of storing different types of checkpointed adaptors, full parameter finetuning might achieve higher performance, as we have verified in our experiments.
>
> **BREAD-LoRA-SGD might yield better results (W3&Q2).** We thank the reviewer for introducing the interesting idea of applying both LoRA and SGD update for inactive blocks. We apply the method in finetuning Llama 3.1-8B under the same setting in Table 1 of the manuscript. The training loss and downstream scores are shown in the following tables:
>
> | Iteration | BAdam  | BREAD-LoRA | BREAD-LoRA+ | BREAD-SGD | BREAD-SGD-LoRA |
> |------|--------|-------------|--------------|------------|-----------------|
> | 3k   | 0.584  | 0.556       | 0.528        | 0.542      | 0.546           |
> | 6k   | 0.562  | 0.523       | 0.493        | 0.515      | 0.507           |
> | 9k   | 0.551  | 0.510       | 0.478        | 0.503      | 0.485           |
> | 12k  | 0.545  | 0.504       | 0.470        | 0.496      | 0.477           |
> | 15k  | 0.541  | 0.500       | 0.467        | 0.492      | 0.473           |
>
> **Table:** Training loss.
>
> | Method             | GSM8K | MATH | NumGLUE | SimulEq | AQuA | Avg.  |
> |--------------------|--------|------|---------|---------|------|--------|
> | BAdam              | 53.9   | 17.2 | 53.7    | 32.5    | 50.4 | 41.5   |
> | BREAD-LoRA         | 57.0   | 20.0 | 55.9    | 32.5    | 49.6 | 43.0   |
> | BREAD-LoRA+        | 57.8   | 20.4 | 56.1    | 32.9    | 51.2 | 43.7   |
> | BREAD-SGD          | 56.9   | 19.6 | 54.1    | 31.5    | 48.0 | 42.0   |
> | BREAD-LoRA-SGD     | 57.3   | 20.1 | 56.0    | 32.7    | 50.4 | 43.3   |
> **Table:** 0-shot score of math finetuning.
>
> | Method             | GSM8K | MATH | NumGLUE | SimulEq | AQuA | Avg.  |
> |--------------------|--------|------|---------|---------|------|--------|
> | BAdam              | 58.3   | 23.6 | 57.2    | 32.8    | 49.6 | 44.3   |
> | BREAD-LoRA         | 57.6   | 23.7 | 58.2    | 32.8    | 50.0 | 44.5   |
> | BREAD-LoRA+        | 61.8   | 24.6 | 58.8    | 32.7    | 51.0 | 45.8   |
> | BREAD-SGD          | 60.6   | 21.4 | 58.2    | 31.8    | 50.8 | 44.6   |
> | BREAD-LoRA-SGD     | 59.2   | 24.0 | 58.4    | 32.6    | 50.6 | 45.0   |
> **Table:** 4-shot score of math finetuning.
>
> We observe that BREAD-LoRA-SGD converges evidently faster than both BREAD-LoRA and BREAD-SGD, which supports the hypothesis that applying both memory-efficient optimization algorithms is better than just applying one. For the downstream task, BREAD-LoRA-SGD also slightly outperforms both BREAD-LoRA and BREAD-SGD in terms of average score. Based on these experiments, applying LoRA+ and SGD might yield better result than BREAD-LoRA+. We leave it as an interesting future research direction. In the next version, we will add the suggested variant and BREAD-SGD-LoRA+ (if this LoRA+ version performs well). Thank you again for the nice suggestion.
>
> **Are the gradient of shallow layers computed?** As shown in equation (5), there are two backward modes. The full backward requires computing the gradient of all the layers, while the efficient backward only computes the gradient up to the active layer.
>
> **Suggestion on the choice of BREAD variants.** BREAD-LoRA variant offers faster convergence and better downstream performance over BREAD-SGD. The observation is consistent with the fact that LoRA can converge faster than SGD for finetuning task, especially when SGD is performed in pure bfloat16 precision. We thereby suggest to use BREAD-LoRA or BREAD-LoRA+ for finetuning tasks.
>
> **Suggestions:**
> * **Acknowledge LOMO.** We will acknowledge and mention LOMO when introducing BREAD-SGD in the next version.
> * **Notation definition.** Thank you for the suggestion. The $s_J^t$ refers to the Adam optimizer states of expansion matrices. We will add a definition in the manuscript.
> * **Discussion on Addax.** We thank the reviewer for referring the related work. Addax applies both first-order and zeroth-order update with different batch sizes, where all the layers are equally treated. In contrast, BREAD distinguishes between active and inactive blocks; the active block drives convergence, while the inactive block facilitates landscape expansion. We will discuss the relationship between BREAD and Addax in the next version of our manuscript.
>
> We hope that our response is satisfactory and addresses all your concerns. If there are any additional questions and concerns, please feel free to let us know during the author-reviewer discussion period. We are more than happy to provide further clarifications.

---

> > ### Comment · Reviewer_FNMe · 2025-08-04
> >
> > I thanks the author for their helpful discussion. I raise my score and I hope the author will include all these changes in revision.

---

> > > ### Author Response · Authors · 2025-08-04
> > > **Thank you for raising the sore**
> > >
> > > We thank the reviewer for raising the score. We are happy to know that our response is helpful and earn your support for acceptance. In the next version of our manuscript, we would include the result of applying multiple light-weight optimizers in landscape expansion (e.g. BREAD-LoRA-SGD) for demonstrating the full potential of the BREAD framework. We would also discuss the relationship between BREAD and Addax/LOMO. We appreciate the reviewer's valuable comments and active participation in the review of our manuscript.

---

### Official Review · Reviewer_rREZ · 2025-07-01

**Clarity:** 3
**Significance:** 3
**Originality:** 3
**Rating:** 4
**Confidence:** 4

**Summary:**

This paper proposes a memory-efficient training method for large language models that addresses limitations of vanilla block coordinate descent (BCD). The authors identify two key issues with applying BCD to neural networks: (1) ineffective utilization of intermediate derivatives during backpropagation, and (2) convergence slowdown due to frozen blocks narrowing the optimization landscape. BREAD addresses these issues by unfreezing inactive blocks and updating them using lightweight methods (LoRA or SGD) during the same backpropagation pass as the active block.

**Questions:**

1. Generalization: The experiments focus primarily on LLaMA models. How well does the approach transfer to other architectures (e.g., MoE, Qwen models)?

2. Statistical significance: The improvements over BAdam are often small (1-3%). It is suggested to provide statistical significance tests across multiple random seeds and the confidence intervals for the reported improvements.

3. How sensitive is the method to the choice of rank r, frequency K, and the ratio between active/inactive learning rates?

**Ethical Concerns:**

["NO or VERY MINOR ethics concerns only"]

**Final Justification:**

I have carefully read the rebuttal and discussion, and my concerns have been addressed. The paper has an interesting discovery, and the proposed method outperforms the SoTA PEFT.

**Limitations:**

Yes

**Quality:**

3

**Strengths And Weaknesses:**

**Strength**
1. The two limitations of BCD for neural networks are clearly articulated and theoretically grounded. The observation that computed gradients for deeper layers are wasted when only updating one block is insightful.
2. Solid theoretical foundation: Propositions 3.1 and 4.1 provide justification for the landscape expansion approach, showing that BCD sub-problems can be strictly suboptimal and that low-rank updates can resolve this issue.
3. Testing on both 8B and 70B models across multiple tasks (math finetuning, instruction tuning, DPO) with thorough ablation studies demonstrates the method's effectiveness.

**Weakness**
1. Limited technical novelty: The core insight of using LoRA for inactive blocks is relatively straightforward. While the theoretical analysis is valuable, the algorithmic contribution feels incremental, essentially combining existing BCD with existing LoRA techniques.
2. No comparison with other recent memory-efficient methods beyond basic baselines (missing comparisons with methods like GaLore, DoRA, etc.)
3. Proposition 3.1 only applies to specific conditions (non-full rank output layer) and may not generalize broadly. The convergence analysis (Theorem 4.2) is informal and based on deterministic gradients, not the stochastic setting used in practice

---

> ### Author Rebuttal · Authors · 2025-07-30
>
> We thank the reviewer for the valuable comments. We now address your concerns below.
>
> **Limited novelty: Using LoRA for inactive block is relatively straightforward.** We would respectfully argue that BREAD is a well-motivated algorithmic *framework*, rather than a trivial combination of BCD and LoRA. The framework naturally incorporates BCD with any light weight optimizers (not just LoRA technique) for alleviating the sub-optimality issue and accelerates the convergence of BCD. Specifically, the effectiveness of BREAD framework has been justified through theory, efficient design, and strong empirical results under tight memory constraints. We list the details below.
>
> *Theoretical insight.* In Proposition 3.1, we identified the suboptimal landscape issue of BCD, where optimizing a single block may lead to suboptimal solution. Proposition 4.2 further reveals that low-rank landscape expansion can mitigate this suboptimality and improve the convergence of BCD. To our knowledge, this phenomenon has not been carefully examined in prior work.
>
> *Computational-efficient algorithmic design.* By leveraging the intermediate derivatives computed during the backpropagation of BCD, BREAD calculates the gradient of the expansion matrices without requiring additional backpropagation steps.
>
> *Competitive performance under limited memory budget.* Similar to LoRA and BAdam, BREAD is able to finetune Llama 3.1-70B model with only A100-80GB GPUs. Compared with memory-efficient baselines, the model finetuned by BREAD attains the highest MT-bench score and average math benchmark score in instruction tuning tasks. Notably, BREAD outperforms Adam in supervised finetuning on Alpaca-GPT4, despite its 80\% less memory consumption.
>
> In summary, we believe that BREAD is a well-motivated algorithmic framework with theoretical insights, computational-efficient design, and competitive empirical performance, rather than a trivial combination of BCD and LoRA.
>
> **No comparison with memory-efficient optimizers such as Galore, DoRA.** We have actually provided the comparison with Galore in terms of memory cost and downstream performance; see our Table 1-3 in the manuscript. To include more methods, we finetune Llama 3.1-8B using DoRA and ReLoRA under the same setting in Table 2 of the manuscript. We present the results in the following two tables, where we also list other methods for completeness.
>
> | Method          | GSM8K | MATH | NumGLUE | SimulEq | AQuA | Avg.  |
> |-----------------|--------|------|---------|---------|------|--------|
> | Base model      | 17.8   | 8.6  | 25.7    | 12.2    | 19.3 | 16.7   |
> | Adam            | 62.3   | 17.4 | 56.4    | 28.6    | 44.9 | 41.9   |
> | Galore          | 46.7   | 16.2 | 42.8    | 28.7    | 47.8 | 36.4   |
> | LoRA-rank80     | 48.7   | 13.7 | 34.6    | 29.6    | 47.3 | 34.8   |
> | DoRA       | 50.5   | 15.0 | 40.2    | 30.0    | 46.1 | 36.4   |
> | ReLoRA          | 47.9   | 18.3 | 49.5    | 31.1    | 45.3 | 38.4   |
> | BAdam           | 53.9   | 17.2 | 53.7    | 32.5    | 50.4 | 41.5   |
> | BREAD-LoRA      | 57.0   | 20.0 | 55.9    | 32.5    | 49.6 | 43.0   |
> | BREAD-LoRA+     | 57.8   | 20.4 | 56.1    | 32.9    | 51.2 | 43.7   |
> | BREAD-SGD       | 56.9   | 19.6 | 54.1    | 31.5    | 48.0 | 42.0   |
> **Table:** 0-shot score of math finetuning.
>
> | Method          | GSM8K | MATH | NumGLUE | SimulEq | AQuA | Avg.  |
> |-----------------|--------|------|---------|---------|------|--------|
> | Base model      | 52.5   | 23.2 | 40.6    | 28.8    | 43.7 | 37.8   |
> | Adam            | 64.9   | 22.9 | 56.8    | 33.5    | 52.8 | 46.2   |
> | Galore          | 57.2   | 22.9 | 45.0    | 32.3    | 48.4 | 41.2   |
> | LoRA-rank80     | 58.1   | 23.0 | 54.4    | 29.0    | 50.3 | 43.0   |
> | DoRA        | 57.2   | 23.1 | 52.5    | 30.9    | 47.6 | 42.3   |
> | ReLoRA          | 55.7   | 24.2 | 56.0    | 31.5    | 48.7 | 43.2   |
> | BAdam           | 58.3   | 23.6 | 57.2    | 32.8    | 49.6 | 44.3   |
> | BREAD-LoRA      | 57.6   | 23.7 | 58.2    | 32.8    | 50.0 | 44.5   |
> | BREAD-LoRA+     | 61.8   | 24.6 | 58.8    | 32.7    | 51.0 | 45.8   |
> | BREAD-SGD       | 60.6   | 21.4 | 58.2    | 31.8    | 50.8 | 44.6   |
> **Table:** 4-shot score of math finetuning.
>
> Notably, both BAdam and BREAD outperform ReLoRA and DoRA. This suggests that BCD-based algorithm might be more suitable for math finetuning task. We will provide a more comprehensive comparison with memory-efficient optimization algorithms in the next version of our manuscript.
>
> **Proposition 3.1 only applies to specific conditions and Theorem 4.2 is for deterministic case.** Let us mention that the primary focus of this work is to design a practical algorithmic framework rather than focusing on theoretical analysis. The derived propositions are mainly intended to motivate the algorithm framework design, and the convergence result serves a partial explanatory purpose to show that the algorithm is partly grounded by classic convergence theory. We consider the convergence under the deterministic setting for compact presentation. We remark that it is possible to combine the current analysis with those of [1, 2] to derive the convergence under stochastic gradient, which we leave as a future work.
>
> **Generalization to different model architectures.** Thank you for the suggestion. We finetune the Qwen 2.5-7B model using the Alpaca-GPT4 dataset for 3 epochs. The MT-bench scores (with GPT-4 as evaluator) are displayed in the following table:
>
> | Model                 | MT-bench Score |
> |-----------------------|----------------|
> | Qwen 2.5-7B (base model) | 7.23          |
> | Galore                | 7.28           |
> | LoRA-rank80           | 7.48           |
> | BAdam                 | 7.49           |
> | BREAD-LoRA            | 7.73           |
> **Table:** MT-bench scores.
>
> The BREAD-LoRA method clearly outperforms baseline methods, which demonstrates the effectiveness of BREAD in finetuning models other than Llama series. We will add these results to include the Qwen model in the next version of our manuscript.
>
> **Statistical significance of the result.** Due to the high computational cost of training LLM, it is almost infeasible to perform multiple runs within the limited rebuttal period. We remark that in Figure 1, BREAD consistently and evidently converges faster than BAdam across different choices of rank and block switch frequency. The consistent faster convergence partially justifies BREAD's performance over BAdam in LLM's finetuning.
>
> **Sensitivity of rank, block switch frequency, and learning rate ratio.** First, he ablation study in Figure 1(c) has shown that BREAD with even rank-1 expansion matrix achieves significantly faster convergence than BAdam. As the rank increases, the convergence speed increases as well. Second, the result in Figure 1(b) shows that BREAD consistently achieves faster convergence as the block switch frequency $K$ increases (under a reasonable number, say 512). This may attribute to that when using larger $K$, the Adam optimizer will aggregate more historical information, leading to a better search direction. Third, we observe that using the same learning rate for the active module and expansion matrices consistently lead to stable convergence for all BREAD variants, and achieve acceleration over BAdam. However, when applying a 10$\times$ larger learning rate in SGD's update of BREAD-SGD, we observe that training will diverge. We conclude that BREAD will safely converge by applying a universal learning rate, and suggest not to assign highly imbalanced learning rate for active and inactive blocks.
>
> We hope that our response is satisfactory and addresses all your concerns. If there are any additional questions and concerns, please feel free to let us know during the author-reviewer discussion period. We are more than happy to provide further clarifications.
>
> **References**
>
> [1] Zhang et al, "Adam can converge without any modification on update rules", NeurIPS 2022.
>
> [2] Li et al, "Convergence of Adam under relaxed assumptions", NeurIPS 2023.

---

> > ### Comment · Reviewer_rREZ · 2025-08-08
> >
> > Thank the authors for the rebuttal and updated results. I will keep my score.

---

> ### Author Response · Authors · 2025-08-09
> **Thank you for your response**
>
> We thank the reviewer for the response. In the next version of our manuscript, we would emphasize more the novelty of our algorithmic framework, provide comparisons with other memory-efficient methods, include experiments on alternative architectures, discuss the assumptions used in our theoretical results, etc. Once again, we appreciate your valuable comments and your active participation in the review of our manuscript.

---

### Official Review · Reviewer_AjsN · 2025-07-02

**Clarity:** 2
**Significance:** 2
**Originality:** 2
**Rating:** 4
**Confidence:** 4

**Summary:**

This paper proposes a novel approach that integrates block coordinate optimization with LoRA-based fine-tuning for large language models. The combination is well-motivated and tackles an important issue: in neural networks, the computational cost of computing gradients for different parameter blocks is not uniform due to the nature of backpropagation, which contrasts with the typical context in which classical coordinate descent methods are applied. The authors highlight this distinction and develop their method specifically to mitigate the resulting inefficiencies in large-scale model training. Experimental results on several benchmarks demonstrate that the proposed methods outperform existing memory-efficient fine-tuning techniques, providing empirical support for the theoretical insights presented in the paper.

**Questions:**

In Figure 1(b), the experiments are limited to K=512. Could the authors clarify why this upper bound was chosen? A broader exploration of K might help identify an optimal value and provide practical guidelines for selecting K in different settings.

**Ethical Concerns:**

["NO or VERY MINOR ethics concerns only"]

**Final Justification:**

Vote for acceptance

**Limitations:**

yes

**Paper Formatting Concerns:**

-

**Quality:**

2

**Strengths And Weaknesses:**

__Strengths__

- **Clear Motivation:** Effectively highlights limitations of classical block coordinate descent for neural network training.
- **Lucid Explanation:** The BREAD framework is explained clearly, with well-presented algorithmic details and connections to existing techniques.
- **Extensive Evaluation:** Presents comprehensive experiments on multiple benchmarks and model sizes, with ablation studies providing insights into design choices.
- **Memory Efficiency:** Demonstrates memory efficiency, a critical consideration for large language model training, with detailed memory cost analysis.
- **Practical Applicability:** Relatively simple to implement and integrate with existing training pipelines.

__Weaknesses__

**Proposition 3.1 (Theory):**
The statement of Proposition 3.1 is unclear regarding the role of $W_{\text{out}}$ on the left-hand side, since minimization is not performed over this variable. Is the strict inequality supposed to hold for all possible $W_{\text{out}}$? This does not seem to be the case, as the inequality would not hold for $W_{\text{out}}^**$ that achieves the minimum in the right-hand side. Therefore, it is not clear for which values of $W_{\text{out}}$ Proposition 3.1 is valid.
I would also appreciate a more detailed discussion following Proposition 3.1. For example, if the result only holds for $W_{\text{out}}$, what about other layers?
Additionally, some variables $W$ in this proposition (and possibly elsewhere in the text) are bolded, while others are not, which may cause confusion.

**Proposition 4.1 (Theory):**
My comments regarding Proposition 4.1 are similar to those for Proposition 3.1. However, while I can at least imagine a proof for Proposition 3.1, Proposition 4.1 appears to be incorrect as stated. For example, if we set $L = 0$ (i.e., there is only $W_{\text{out}}$) and $W_{\text{out}} = 0$ on the left-hand side, Proposition 4.1 claims that minimization over a rank-1 matrix is equivalent to minimization over the full matrix $W_{\text{out}}$, which is clearly not the case. It is essential to provide a precise explanation of the conditions under which Proposition 4.1 holds, or to clarify its statement.
Furthermore, the proof in Appendix D.2 is very brief (only three lines) and does not provide enough detail to clarify the underlying reasoning, making it difficult to assess the validity of the propositions.

**Theorem 4.2 (Theory):**
First, Theorem 4.2 appears to be more of an intermediate result (a descent lemma) rather than a final convergence rate. Typically, at this stage, one would sum over $k$ so that all $H(\cdot)$ terms except the first and last cancel out, leading to a convergence rate of the form $O(1/\sqrt{K})$. However, in the BREAD algorithm, the relevant convergence should be with respect to $t$, not $K$, since the number of steps per block is not very large (in your experiments, as I understand, it was at most 512). As a result, the theorem does not actually provide a meaningful convergence guarantee for the overall algorithm.
I also do not fully understand why classical convergence results for Adam cannot be applied here, since $t$ is not changing. Similarly, it is unclear why standard results for coordinate descent cannot be used.
Finally, Theorem 4.2 is stated for deterministic (full-batch) gradients, which is not realistic for LLM training tasks, where stochastic gradients are always used.

**Typos and Minor Issues**
- "landscape expansionutilizes" (should be "landscape expansion utilizes")
- "landscape expansioncomponent" (should be "landscape expansion component")
- "landscape expansiontechnique" (should be "landscape expansion in")
- "landscape expansionapproaches" (should be "landscape expansion approaches")
- "simpliest implementaiton" (should be "simplest implementation")
- "requries" (should be "requires")
- "LoRA’ variants" (should be "LoRA variants" or "LoRA's variants")
- The term "block-epoch" is sometimes written as "block epoch"; it is preferable to use a consistent form throughout the manuscript.

This work primarily focuses on the proposed algorithmic framework and its empirical evaluation, while some theoretical aspects and broader experimental validation remain open for future research. In particular, the convergence analysis is limited in scope, and the experiments are conducted on selected tasks and models.

I borderline reject the paper for now, but I look forward to communicating with the authors.

---

> ### Author Rebuttal · Authors · 2025-07-29
>
> We thank the reviewer for the valuable comments. Overall, let us mention that our primary focus lies in proposing the BREAD algorithmic framework (and its practical implementation) that accelerates BCD for LLM finetuning, rather than focusing on theoretical analysis. The derived propositions are mainly intended to motivate the algorithm design, and the convergence result serves a partial explanatory purpose to show that the algorithm is grounded by classic convergence theory. Below, we address your concerns regarding our theoretical results.
>
> **Confusion in Proposition 3.1.** The objective of Proposition 3.1 is to show that fixing $W_{\text{out}}$ may potentially exclude the optima, which can happen in the BCD update scheme. As stated in Proposition 3.1, we require $W_{\text{out}}$ in the left hand side to be not full column rank for the strict inequality to hold. Namely, if one fixes $W_{\text{out}}$ at a degenerate point, optimizing other variables only (such as the scheme of BCD) will attain a strictly higher function value compared to the optimal value (i.e., the left hand side). Then, based on this result we conjecture that this issue may potentially slow down the convergence of BCD's subproblem, as it may not include the optima at all. This observation motivates the design of the BREAD algorithmic framework.
>
> Since the feature learning depends on the model parameterization, characterizing the sub-optimality of BCD's subproblem for other layers (other than the last layer) for an *arbitrary* model is almost impossible. Instead, let us use the following much simplified regression model to demonstrate the effect of fixing one of the other layers. Since the sub-optimality holds for such a simple model, we would then extrapolate that the same issue will likely remain for the training of the much more complicated language models.
>
> Consider the following regression problem
> $$\min_{W} ||H(W, X) - Y||\_2^2$$
> where the input is $X \in \mathbb{R}^{B\times m}$ and the ground truth label $Y=PX \in \mathbb{R}^{B\times n}$, with rank($P$)=$r$. Suppose H(W, X) is a multi-layer model:
> $$H(W, X):= f_{W_L} \circ \cdots f_{W_2} \circ f_{W_1}(X)$$
> $f_{W_\ell}(Z):=\sigma(W_\ell Z)$,
> where $\sigma(\cdot)$ is the ReLU activation function. Suppose one fix any $W_\ell$ with rank($W_\ell$)<r, then by the rank inequality there always exists $(X, Y$) such that
> $$\min_{\{W \backslash W_\ell\}} ||H(W, X) - Y||\_2^2 > \min_{W} ||H(W, X) - Y||\_2^2$$
>
> In addition, we will change all the $W$ in the paper to boldface to keep consistency.
>
> **Proposition 4.1’s validity.** When L=0, the problem essentially becomes finding $Cx=e_y$. One rank-1 solution is let $C_{:, i}=0 \ \forall i \neq y$, $C_{:, y}=\frac{x}{||x||_2^2}$. Therefore, Proposition 4.1 correctly holds under the scenario mentioned in the review.
>
> **Clarification on Theorem 4.2.** The sample complexity is established by telescoping the inequality over block epoch $t$ rather than the block subproblem step $K$. We now sketch the derivation of the sample complexity based on the inequality of Theorem 4.2. Let us recall the following definitions:
>
> - $W^t$: Weight at the start of block epoch $t$.
> - $W_i^t$: Weight  at the start of block epoch $t$, block sub-problem $i$.
> - $W_{i}^{t, k}$: Weight at block epoch $t$, block sub-problem $i$, update step $k$.
>
> The Theorem 4.2 gives
> $$H(W_{i}^{t, k+1})-H(W_{i}^{t, k}) \leq \mathcal{O}(\alpha) ||\nabla_i H(W_{i}^{t,k})||\_2^2 - \mathcal{O}(\beta) \sum_j ||\nabla_{C_j} H(W_i^{t,k})||\_2^2$$
> Sum over $k=1, \cdots, K$ steps and apply the assumption of smoothness and bounded gradient gives
> $$H(W_{i+1}^t) - H(W_{i}^{t}) \leq -K \mathcal{O}(\alpha) ||\nabla_i H(W_i^t)||\_2^2 - K \mathcal{O}(\beta) \sum_j ||\nabla_{C_j} H(W_i^t)||\_2^2 + \mathcal{O}(K^2(\alpha^3+\alpha^2\beta))$$
> Sum over blocks $i=1, \cdots, D$ gives
> $$H(W^{t+1}) - H(W^t)$$
> $$\leq-K\mathcal{O}(\alpha) \sum_{i=1}^{D} ||\nabla_i H(W_i^t)||\_2^2 - DK\mathcal{O}(\beta) \sum_j ||\nabla_{C_j} H(W_i^t)||\_2^2 + \mathcal{O}(DK^2(\alpha^3+\alpha^2\beta))$$
> $$\leq -K\mathcal{O}(\alpha)||\nabla_W H(W^t)||\_2^2 - DK\mathcal{O}(\beta) ||\nabla_C H(W^t)||\_2^2 + \mathcal{O}(D^2K^2(\alpha^3+\alpha^2\beta)),$$
> where the last inequality follows from the assumption of smoothness and bounded gradient.
>
> Let $\alpha=\mathcal{O}(1/\sqrt{T}), \beta=\mathcal{O}(1/\sqrt{T})$ and telescope the inequality over $t=1, \cdots, T$, we have
> $$\frac{1}{T} \sum_{t=1}^{T}||\nabla_W H(W^t)||_2^2 = \mathcal{O}(\frac{1}{\sqrt{T}K} + \frac{K^2}{T^{1.5}}).$$
>
> We will derive the detailed convergence rate in the next version of our manuscript.
>
> **Why classical Adam's convergence result cannot be applied, since $t$ is not changing.** We clarify that $t$ is the block epoch and changes over training. While one can follow the Adam's analysis to analyze the descent of one block subproblem, the gradient norm of the converged point will contain constant error of $\mathcal{O}(1/\sqrt{K})$, as $K$ is a prefixed constant value. To establish the convergence of the algorithm, one needs to jointly analyze the descent of a full block epoch rather than a single block sub-problem.
>
> **Why standard coordinate descent result cannot be used?** The update of the active block uses Adam update rule rather than gradient descent update. Therefore, the bias between Adam update and the true block gradient has to be bounded using a different analysis; see equation 12-14 in Appendix D for the error analysis of Adam update.
>
> **The convergence result is for deterministic case.** Since our main focus for this work is to design a practical algorithmic framework rather than establishing new theoretical convergence result, we consider the convergence under the deterministic setting for compact presentation. We remark that it is possible to combine the current analysis with [1, 2] to derive the convergence under stochastic gradient, which we leave as a future work.
>
> **Typos.** We thank the reviewer for the careful reading of our manuscript, and will revise all the typos in the next version of our manuscript.
>
> **Choice of K beyond 512.** Our preliminary experiments show that once $K$ reaches above 512, the convergence acceleration becomes marginal. Hence, we only present the convergence result up to $K=512$. This may due to that the algorithm will overly optimize one block before moving to the next block and slow down the convergence.
>
> We hope that our response is satisfactory and addresses all your concerns. If there are any additional questions and concerns, please feel free to let us know during the author-reviewer discussion period. We are more than happy to provide further clarifications.
>
> **References**
>
> [1] Zhang et al, "Adam can converge without any modification on update rules", NeurIPS 2022.
>
> [2] Li et al, "Convergence of Adam under relaxed assumptions", NeurIPS 2023.

---

> > ### Comment · Reviewer_AjsN · 2025-08-04
> >
> > Thank you so much for your reply! I raised my score. I recommend that the authors take into account the changes from the reply.

---

> ### Author Response · Authors · 2025-08-04
> **Thank you for raising the score**
>
> We thank the reviewer for acknowledging our response and raising the score. In the next version of our manuscript, we would include more detailed explanation on Proposition 3.1, and discuss the scenarios where the strict sub-optimality holds when intermediate layers are frozen. We would also rewrite the convergence statement, provide detailed derivation for the sample complexity, and discuss the challenge of our analysis compared to previous works. Additionally, we will correct all the typos you noted. We appreciate the reviewer's valuable comment and active participation in the review of our manuscript.

---

### Decision · Program_Chairs · 2025-09-17

**Decision:**

Accept (poster)

**Comment:**

This paper presents BREAD, a memory-efficient fine-tuning algorithm that enhances Block Coordinate Descent (BCD) by addressing wasted computation and limited optimization landscapes. By updating inactive blocks with lightweight methods during the same backpropagation pass, BREAD achieves better optimization with minimal overhead.

The rebuttal addressed reviewers' concerns, and the method demonstrates strong empirical performance, outperfoming state-of-the-art PEFT approaches. Overall, this is a well-motivated paper, therefore I recommend acceptance.